# Learning prediction error neurons in a canonical interneuron circuit

**Loreen Hertäg[1,2]\*, Henning Sprekeler[1,2]\***

[1]Modelling of Cognitive Processes, Institute of Software Engineering and Theoretical Computer Science, Berlin Institute of Technology, Berlin, Germany; [2]Bernstein Center for Computational Neuroscience, Berlin, Germany

**Abstract** Sensory systems constantly compare external sensory information with internally generated predictions. While neural hallmarks of prediction errors have been found throughout the brain, the circuit-level mechanisms that underlie their computation are still largely unknown. Here, we show that a well-orchestrated interplay of three interneuron types shapes the development and refinement of negative prediction-error neurons in a computational model of mouse primary visual cortex. By balancing excitation and inhibition in multiple pathways, experience-dependent inhibitory plasticity can generate different variants of prediction-error circuits, which can be distinguished by simulated optogenetic experiments. The experience-dependence of the model circuit is consistent with that of negative prediction-error circuits in layer 2/3 of mouse primary visual cortex. Our model makes a range of testable predictions that may shed light on the circuitry underlying the neural computation of prediction errors.

**\*For correspondence:**
loreen.hertaeg@tu-berlin.de (LHä);
h.sprekeler@tu-berlin.de (HS)

**Competing interests:** The authors declare that no competing interests exist.

## Introduction

Changes in sensory inputs can arise from changes in our environment, but also from our own movements. When you walk through a room full of people, your perspective changes over time, and you will experience a global visual flow. Superimposed on this global change are local changes generated by the movements of the people around you. An essential task of sensory perception is to disentangle these different origins of sensory inputs, because the appropriate behavioral responses to environmental and to self-generated changes are often different. Am I approaching a person or is she approaching me?

A common assumption is that perceptual systems subtract from the sensory data an internal prediction (*Bell, 1981*; *Rao and Ballard, 1999*; *Friston, 2005*; *Spratling, 2010*; *Franklin and Wolpert, 2011*; *den Ouden et al., 2012*; *Kennedy et al., 2014*; *Keller and Mrsic-Flogel, 2018*), which is calculated from an efference copy of the motor signals our brain has issued. Changes in the external world then take the form of mismatches – or prediction errors – between internal predictions and sensory data (*Wolpert et al., 1995*). This comparison requires an accurate prediction system that adapts to ongoing changes in the environment or in behavior. An efficient way to ensure a flexible adaptation is to render the prediction circuits experience-dependent by minimizing prediction errors (*Wolpert et al., 2011*).

Neural hallmarks of prediction errors are found throughout the brain. Dopaminergic neurons in the basal ganglia and the striatum (*Schultz and Dickinson, 2000*) encode a reward prediction error (mismatch between expected and received reward), and subsets of neurons in visual cortex (*Keller et al., 2012*; *Zmarz and Keller, 2016*; *Attinger et al., 2017*), auditory cortex (*Eliades and Wang, 2008*; *Keller and Hahnloser, 2009*) and barrel cortex (*Ayaz et al., 2019*) code for a mismatch between feedback and feedforward information.

While neural correlates of prediction errors have been found broadly, the circuit level mechanisms that underlie their computation are poorly understood. Given that prediction errors involve a

subtraction of expectations from sensory data (for an alternative implementation employing divisive inhibition, see *Spratling, 2008*; *Spratling, 2017*; *Spratling, 2019*), the relevant circuits likely involve both excitatory and inhibitory pathways (*Attinger et al., 2017*). Negative prediction-error (nPE) neurons, which are activated only when sensory signals are weaker than predicted, are likely to receive excitatory predictions counterbalanced by inhibitory sensory signals. Conversely, positive prediction-error (pPE) neurons, which respond only when sensory signals exceed the internal prediction, may receive excitatory sensory signals counterbalanced by inhibitory predictions (*Rao and Ballard, 1999*; *Keller and Mrsic-Flogel, 2018*). How the complex inhibitory circuits of the cortex (*Markram et al., 2004*; *Rudy et al., 2011*; *Pfeffer et al., 2013*; *Jiang et al., 2015*; *Tremblay et al., 2016*; *Wamsley and Fishell, 2017*) support the computations of these prediction errors is not resolved and neither are the activity-dependent forms of plasticity that would allow these circuits to refine the prediction machine.

For prediction-error neurons, fully predicted sensory signals should cancel with the internal prediction and hence trigger no response. We therefore hypothesized that an experience-dependent formation and refinement of prediction-error circuits can be achieved by balancing excitation and inhibition in an activity-dependent manner. Using a computational model comprised of excitatory pyramidal cells and three types of inhibitory interneurons, we show that nPE neurons can be learned by inhibitory synaptic plasticity rules that balance excitation and inhibition in principal cells. We find that the circuit shows a similar experience dependence as observed in V1 (*Attinger et al., 2017*). Depending on which interneuron classes receive motor predictions and which receive sensory signals, the plasticity rules shape different, fully functional variants of the prediction circuit. Using simulated optogenetic experiments, we show that these variants have identifiable fingerprints in their reaction to optogenetic activation or inactivation of different interneuron classes. Finally, we demonstrate that the inhibitory prediction circuits can be learned by biologically plausible forms of homeostatic inhibitory synaptic plasticity, which only rely on local information available at the synapses.

## Results

We studied a rate-based network model of layer 2/3 of rodent V1 to investigate how prediction-error (PE) neurons develop. In the following, we will focus primarily on negative prediction-error (nPE) neurons. In V1, nPE neurons have been studied more extensively, which allows us to compare our results with experimental findings. However, the same approaches and principles derived for nPE neurons can also be applied to positive prediction-error (pPE) neurons (see *Appendix 2—figures 1* and *2*). The network model includes excitatory pyramidal cells (PCs) as well as inhibitory parvalbumin-expressing (PV), somatostatin-expressing (SOM) and vasoactive intestinal peptide-expressing (VIP) interneurons (*Figure 1a*). The relative abundance of the four cell types and the probability of the respective synaptic connections are chosen in line with electrophysiological studies (see Materials and methods). While all inhibitory neurons are modeled as point neurons (*Wilson and Cowan, 1972*), we used a two-compartment model for PCs with a rectifying active dendritic process that allowed nonlinear dynamics akin to dendritic calcium spikes (*Murayama et al., 2009*) (see Materials and methods and Appendix 1).

A subset of inhibitory synapses – chosen based on a mathematical analysis (see Materials and methods, or Appendix 1) – are subject to experience-dependent plasticity, which aims at minimizing deviations of the PC firing rate from a baseline rate. These deviations can be interpreted as prediction errors. Learning hence strives to adapt the inhibitory circuit such as to reduce these errors. While the synapses onto both the somatic and dendritic compartments of PCs follow an inhibitory plasticity rule akin to *Vogels et al., 2011*, the inhibitory synapses onto PV neurons follow an approximated backpropagation of error rule akin to *Rumelhart et al., 1986*. Specifically, the former rule changes the synapses onto PCs in proportion to the presynaptic interneuron activity and the deviation of PC activity from a baseline rate (see Materials and methods, *Equation 14*). The latter rule changes the synapses onto PV neurons in proportion to the presynaptic interneuron activity and the averaged deviation of the postsynaptic PCs from their baseline rate (see Materials and methods, *Equation 16*). Earlier work has shown that such forms of plasticity establish a balance of excitation and inhibition (*Vogels et al., 2011*; *Mackwood et al., 2020*).

All neurons in the model receive excitatory background input that ensures reasonable baseline activities in the absence of visual input and motor-related internal predictions ('baseline'). In

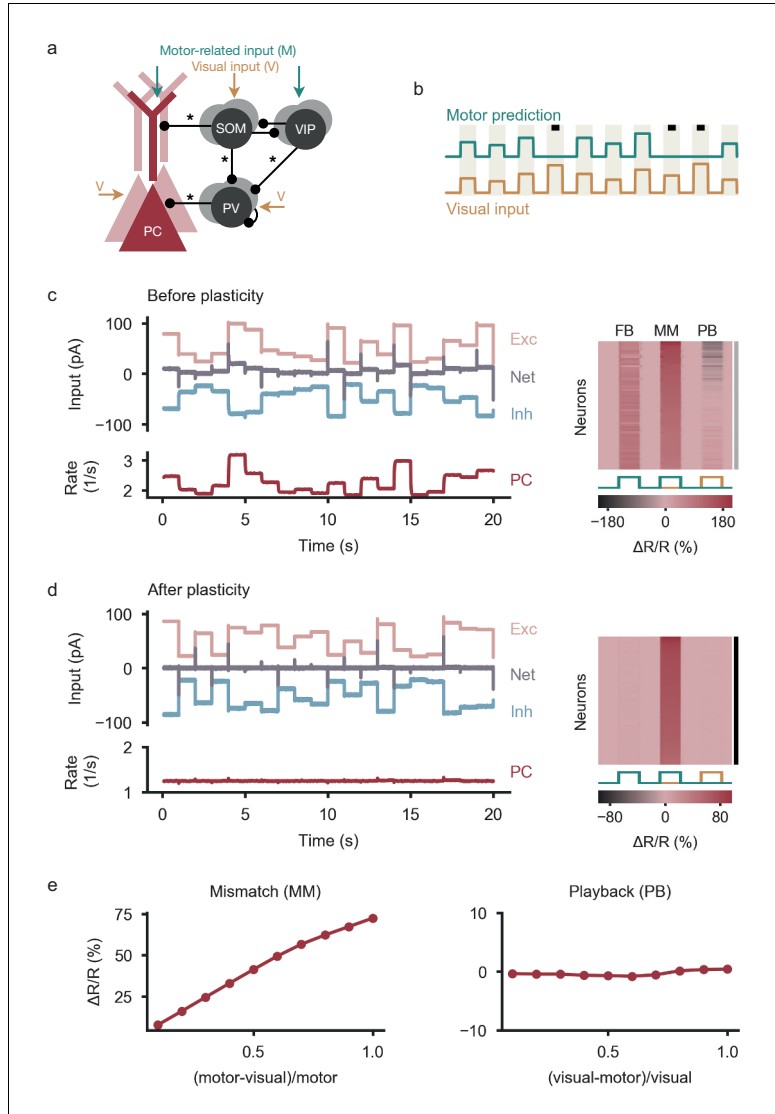

**Figure 1.** Balancing excitation and inhibition gives rise to negative prediction-error neurons. (**a**) Network model with excitatory PCs and inhibitory PV, SOM and VIP neurons. Connections from PCs not shown for the sake of clarity. Somatic compartment of PCs, SOM and PV neurons receive visual input, apical dendrites of PCs and VIP neurons receive a motor-related prediction thereof. Connections marked with an asterisk undergo experience-dependent plasticity. (**b**) During plasticity, the network is exposed to a sequence of feedback (coupled sensorimotor experience) and playback phases (black square, visual input not predicted by motor commands). Stimuli last for 1 s and are alternated with baseline phases (absence of visual input and motor predictions). (**c**) Left: Before plasticity, somatic excitation (light red) and inhibition (light blue) in PCs are not balanced. Excitatory and inhibitory currents shifted by ±20 pA for visualization. The varying net excitatory current (gray) causes the PC population rate to deviate from baseline. Right: Response relative to baseline ($\Delta R/R$) of all PCs in feedback (FB), mismatch (MM) and playback (PB) phase, sorted by amplitude of mismatch response. None of the PCs are classified as nPE neurons (indicated by gray shading to the right). (**d**) Same as in (**c**) after plasticity. Somatic excitation and inhibition are balanced. PC population rate remains at baseline. All PCs classified as nPE neurons (also indicated by black shading to the right). (**e**) Left: Mismatch response increases with the difference between visual and motor input. Right: nPE neuron response during playback does not change with the difference between visual and motor input but remains at baseline.

The online version of this article includes the following figure supplement(s) for figure 1:

**Figure supplement 1.** Learning prediction-error circuits with different forms of homeostatic plasticity.

**Figure supplement 2.** VIP→PV synapses are not required for the formation of nPE neurons.

*Figure 1 continued on next page*

*Figure 1 continued*

**Figure supplement 3.** Balancing excitation, somatic and dendritic inhibition gives rise to nPE neurons in a model in which an excess of dendritic inhibition is forwarded to the soma.

addition, we stimulated the network with time-varying external inputs representing actual and predicted visual stimuli (*Figure 1a,b*). We reasoned that during natural conditions, movements lead to sensory inputs that are fully predicted by internal motor commands ('feedback phase', *Attinger et al., 2017*), while unexpected external changes in the environment should generate unpredicted sensory signals ('playback phase', *Attinger et al., 2017*). Situations in which internal motor commands are not accompanied by corresponding sensory signals should be rare ('feedback mismatch phase', *Attinger et al., 2017*). During plasticity, we therefore stimulated the circuit with a sequence consisting of feedback and playback phases ('quasi-natural training', *Figure 1b*).

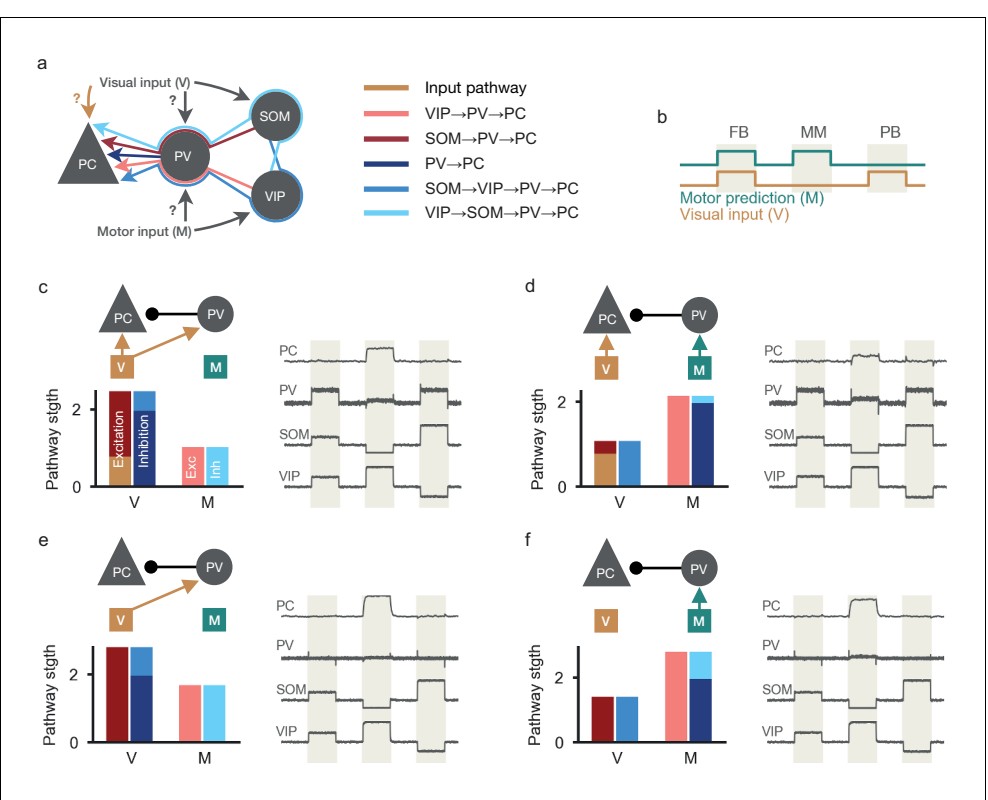

**Figure 2.** Multi-pathway balance of excitation and inhibition in different nPE neuron circuits. (**a**) Excitatory, inhibitory, disinhibitory and dis-disinhibitory pathways onto PCs that need to be balanced in nPE neuron circuits. Input to the soma of PCs and PV neurons is varied (**c**–**f**). SOM neurons receive visual input, VIP neurons receive a motor-related prediction. (**b**) Test stimuli: Feedback (FB), mismatch (MM) and playback (PB) phases of 1 s each. (**c**) PCs and PV neurons receive visual input (left, top). When all visual (V) and motor (M) pathways are balanced (left, bottom), PCs act as nPE neurons (right). PV neuron activity increases in both feedback and playback phases. Responses normalized between −1 and 1 such that baseline is zero. (**d**) Same as in (**c**) but PV neurons receive motor predictions. (**e**) Same as in (**c**) but PCs receive no visual input. PV neurons remain at baseline in the absence of visual input to the soma of PCs. (**f**) Same as in (**c**) but PCs receive no visual input and PV neurons receive motor predictions. PV neurons remain at baseline in the absence of visual input to the soma of PCs.
The online version of this article includes the following figure supplement(s) for figure 2:

**Figure supplement 1.** Multi-pathway balance of excitation and inhibition in different nPE neuron circuits with both visual and motor input onto PV neurons.

## Negative prediction-error neurons emerge by balancing excitation and inhibition

Before the onset of plasticity, synaptic connections were randomly initialized, leading to PCs receiving unbalanced excitation and inhibition. Therefore, all PCs change their firing rate in response to both feedback and playback stimuli, indicating the absence of nPE neurons (*Figure 1c*). During quasi-natural sensorimotor experience, inhibitory plasticity strengthens or weakens inhibitory synapses to diminish the firing rate deviations of PCs from their baseline firing rate (*Figure 1—figure supplement 1*). At the same time, dendritic inhibition mediated by SOM interneurons was sufficiently strengthened to suppress the motor prediction arriving at the apical dendrite. After synaptic plasticity, somatic excitation and inhibition are balanced on a stimulus-by-stimulus basis (*Figure 1d–e*). PCs merely show small and transient onset/offset responses to feedback and playback stimuli. In contrast, all PCs show an increase in activity for feedback mismatch stimuli (*Figure 1d*), which scales with the size of the difference between actual and predicted visual input (*Figure 1e*). Hence, inhibitory synaptic plasticity generates nPE neurons by balancing excitation and inhibition in PCs for quasi-natural conditions.

## Balance of excitation, inhibition and disinhibition in different functional prediction circuits

The circuit we studied so far was motivated by the widely accepted view that PCs and SOM and PV interneurons show visual responses (*Ko et al., 2011*; *Yang et al., 2013*; *Larkum, 2013a*; *Xue et al., 2014*; *Harris and Shepherd, 2015*; *Lee et al., 2016*; *Attinger et al., 2017*), while long-range (motor) predictions arrive in the superficial layers of V1 and target VIP neurons (*Fu et al., 2014*; *Harris and Shepherd, 2015*; *Tremblay et al., 2016*; *Attinger et al., 2017*) and the apical and distal compartments of PCs (*Larkum, 2013a*; *Leinweber et al., 2017*; *Attinger et al., 2017*). Because this view is not uncontested (*Fu et al., 2014*) and it has been shown that interneuron types can receive both feedforward and feedback inputs (*Wall et al., 2016*), we systematically varied the inputs to the different neuron classes. We first studied circuit variations in which PCs and PV neurons receive visual and/or motor signals (*Figure 2*, see also *Figure 2—figure supplement 1*).

We found that inhibitory plasticity establishes nPE neurons independent of the input configuration onto PCs and PV neurons (*Figure 2c–f*, right). The emerging connectivity of the interneuron circuits varied, however. For PCs not to respond above baseline in feedback and playback phase, various excitatory, inhibitory, disinhibitory and dis-disinhibitory pathways need to be balanced. An informative example is the input configuration in which PCs receive visual input and PV neurons receive motor predictions (*Figure 2d*). In this case, visual inputs arrive at the PCs as direct excitation, as disinhibition through the SOM-PV pathway, and as dis-disinhibition via the SOM-VIP-PV pathway (*Figure 2a*). To keep the PCs at their baseline during the playback phase, these three pathways need to be balanced (*Figure 2d*, left). Similarly, motor signals arrive at the PCs as inhibition from PV neurons, dis-inhibition via the VIP-PV pathway, dis-dis-inhibition via the VIP-SOM-PV pathway and as direct excitation to the dendrite that is canceled by SOM-mediated inhibition. Again, all these pathways need to be balanced to keep the PCs at their baseline for fully predicted visual stimuli (*Figure 2d*, left). Analog balancing arguments hold for other input configurations (*Figure 2c–f*, left). Note that this multi-pathway balance applies primarily to somatic inputs to PCs. During feedback and playback phases, this provides a complete picture, because the dendrites are deactivated by inhibition. During mismatch phases, this dendritic inhibition is withdrawn and the dendrites provide additional excitatory input to the soma that can drive mismatch responses.

While the flow of visual and motor information in the learned inhibitory microcircuit is different for different input configurations, the neural responses of the different interneuron classes provide limited information about the input configuration. PV neuron activity reflects whether PCs receive visual input: If PCs receive visual input, PV responses increase during feedback and playback phases to balance the sensory input at the soma of PCs (*Figure 2c–d*, right). If PCs receive no visual input, PV neurons remain at their baseline firing rate (*Figure 2e–f*, right), which is in contradiction to the experimentally observed increase of PV neurons during feedback (see *Attinger et al., 2017*). The activity of SOM and VIP neurons varies between playback, feedback and mismatch phases (in line with experimental results, see *Attinger et al., 2017*), but is independent of the input configuration for PCs and PV interneurons (*Figure 2c–f*, right).

In summary, inhibitory plasticity can establish functional nPE circuits irrespective of the inputs onto the soma of PCs and PV neurons. Although the underlying circuits vary substantially in the specific balance of pathways, the neural activity patterns only weakly reflect the underlying information flow.

## Simulated optogenetic manipulations disambiguate prediction-error circuits

We hypothesized that the need to simultaneously balance several pathways offers a way to disambiguate the different prediction-error circuits by optogenetic manipulations. To test this, we systematically suppressed or activated PV, SOM and VIP interneurons in each input configuration after inhibitory plasticity had established the respective nPE circuit.

We found that in our model, such simulated optogenetic experiments are highly informative about the underlying input configuration (*Figure 3*). For example, PV neuron inactivation changes the response of nPE neurons during feedback, playback and mismatch phases if and only if the PCs receive visual inputs. VIP inactivation renders nPE neurons silent unless PV neurons receive motor predictions, in which case they are transformed into positive prediction-error (pPE) neurons. Since

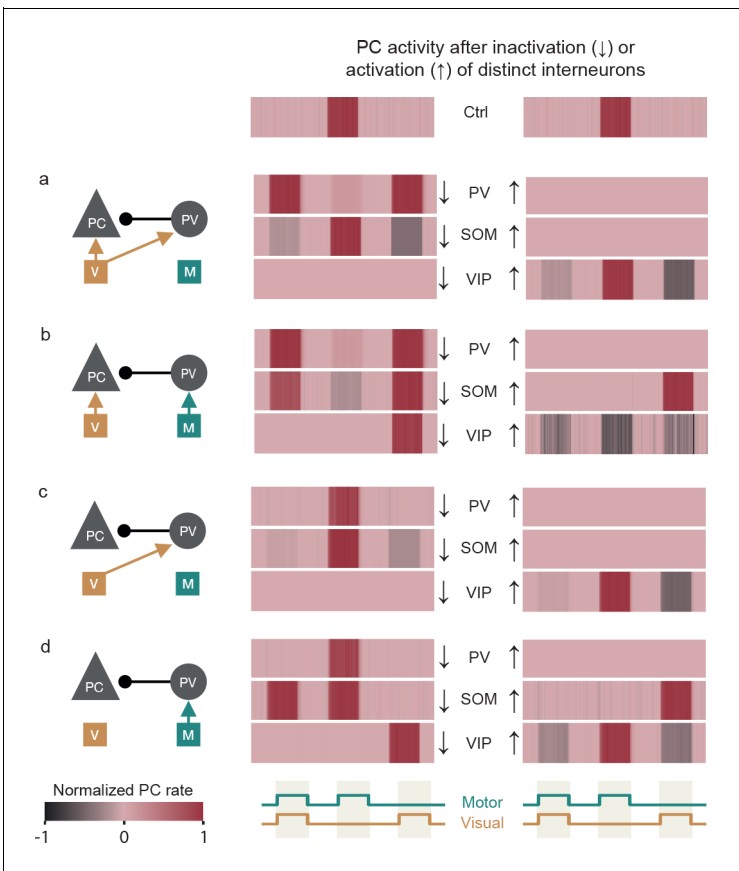

**Figure 3.** Simulated optogenetic manipulations of PV, SOM and VIP neurons disambiguate prediction-error circuits. (a) Left: nPE neuron circuit in which PCs and PV neurons receive visual input. Inactivation (middle) or activation (right) of PV (first row), SOM (second row) or VIP neurons (third row). Optogenetic manipulations change responses of nPE neurons (Ctrl) in feedback, mismatch and playback phases. Responses normalized between −1 and 1 such that baseline is zero. Inactivation input is -8 $s^{-1}$. Activation input is 5 $s^{-1}$. (b) Same as in (a) but PV neurons receive motor-related prediction. (c) Same as in (a) but PCs receive no visual input. (d) Same as in (a) but PCs receive no visual input and PV neurons receive a motor-related prediction.

The online version of this article includes the following figure supplement(s) for figure 3:

**Figure supplement 1.** Net currents in PCs after in/activation of PV, SOM or VIP neurons elucidate prediction-error circuits.

SOM and VIP neurons are mutually inhibiting (see e.g. *Pfeffer et al., 2013*), the same information can be gained by an over-activation of SOM neurons that effectively silences VIP neurons.

Changes in neuronal activity due to optogenetic manipulations depend on a variety of factors such as baseline firing rates and saturation effects (*Phillips and Hasenstaub, 2016*). For instance, while an excess of inhibition is not observable when PCs exhibit vanishingly small baseline activity, it leads to a firing rate decrease otherwise. Moreover, in/activation of interneuron types within a recurrent network may also have ambiguous consequences contingent on potential saturation effects in other cell types. These ambiguities can be partially resolved by measuring currents rather than firing rates, during baseline, feedback, mismatch and playback phases. Indeed, we found that the net currents in PCs after in/activation of PV, SOM or VIP neurons are highly informative about the underlying input configuration (*Figure 3—figure supplement 1*).

When we compared our results with optogenetic experiments in which SOM or VIP neurons are either inactivated or activated during mismatch or running (*Attinger et al., 2017*), it shows that a homogeneous input configuration in which all PCs receive visual input while all PV neurons receive a motor-related prediction thereof is unlikely (*Figure 3b*). All other variants of nPE circuits exhibit mismatch responses during SOM/VIP neuron manipulation that are in line with the ones observed experimentally. However, the responses observed in the feedback phase (when compared with 'during running', see *Attinger et al., 2017*) deviate from all the conditions we simulated, indicating that the interneurons do not receive exclusively sensory or motor inputs, but rather a combination of actual and predicted visual input.

In summary, our model predicts that optogenetic experiments may unveil a unique fingerprint for nPE circuits that differ in their inputs onto PCs and PV neurons.

## Fraction of nPE neurons is modulated by inputs to SOM and VIP interneurons

In the model considered so far, all PCs developed into nPE neurons during learning, irrespective of the inputs to PCs and PV interneurons. However, nPE neurons represent only a small fraction of neurons in mouse V1 (*Keller et al., 2012*; *Saleem et al., 2013*; *Zmarz and Keller, 2016*; *Attinger et al., 2017*). Given that in our model, motor predictions arriving at the apical dendrites are canceled by SOM neuron-mediated inhibition, we hypothesized that the fraction of PCs that develop into nPE neurons depends on the distribution of visual and motor input onto SOM and VIP neurons.

To test this, we allow neurons of both SOM and VIP populations to receive either visual input or a motor prediction thereof. A fraction $f$ of SOM neurons and a fraction $(1-f)$ of VIP neurons receive

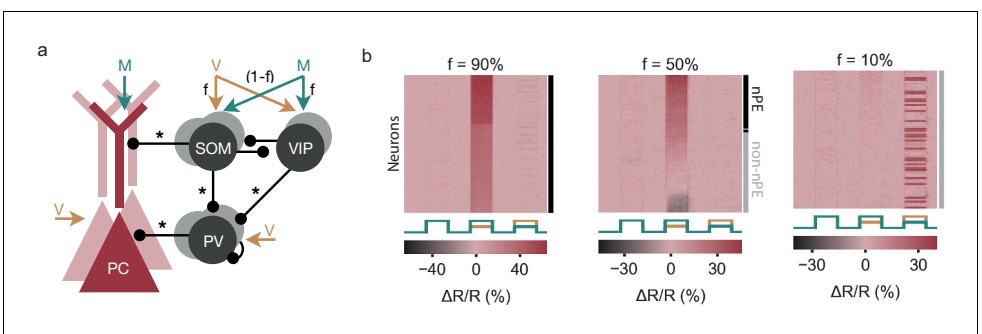

**Figure 4.** Fraction of nPE neurons depends on SOM and VIP neuron inputs. (a) Network model with excitatory PCs and inhibitory PV, SOM and VIP neurons. Connections from PCs not shown for the sake of clarity. Somatic compartment of PCs, PV neurons, a fraction $f$ of SOM neurons and a fraction $(1-f)$ of VIP neurons receive visual input. The remaining SOM and VIP neurons receive motor predictions. (b) Response relative to baseline ($\Delta R/R$) of all PCs in feedback, mismatch and playback phases, sorted by amplitude of mismatch response. The fraction of nPE neurons that develop during learning decreases with $f$ (also indicated by black and gray shading to the right). The increasing fraction of non-nPE neurons comprises neurons that remain at their baseline in all three phases, show a suppression during mismatch or develop into positive prediction-error neurons that respond only during playback.

visual input. The remaining SOM and VIP neurons receive a motor-related prediction (*Figure 4a*). When the majority of SOM neurons receive visual inputs and the majority of VIP neurons receive motor predictions ($f \approx 1$), all PCs develop into nPE neurons (*Figure 4b*, left). Reducing the proportion of SOM neurons that receive visual input (and, equivalently, the proportion of VIP neurons that receive the motor prediction), the fraction of nPE neurons decreases (*Figure 4b*, middle). Non-nPE neurons remain at their baseline in all three phases, show a suppression during mismatch or develop into pPE neurons that respond only during playback. pPE neurons only emerge when the inputs to SOM and VIP neurons are reversed such that most SOM neurons receive motor predictions (*Figure 4b*, right).

In summary, the fraction of nPE neurons that develop during learning depends on the distribution of visual input and motor predictions onto both SOM and VIP neurons.

## Experience-dependence of mismatch and interneuron responses

*Attinger et al., 2017* showed that the number of nPE neurons and the strength of their mismatch responses decreases when mice are trained in artificial conditions, during which a mouse was shown the visual information of a different mouse, such that motor predictions and visual flow were uncorrelated ('non-coupled training'). We reasoned that this training paradigm should include baseline phases where both animals sit still and phases, during which the speeds of the two animals differ. To test whether the model shows the same experience-dependence, we generated a modified training paradigm, which includes baseline phases and phases during which the visual inputs and motor-related predictions are statistically independent ('random gain training', *Figure 5a*). We found that the number of nPE neurons and their mismatch responses also decrease for random gain trained relative to quasi-natural trained networks (*Figure 5b*). This decrease is primarily due to changes in PCs and PV neurons, while the responses of SOM and VIP neurons during the mismatch phase are largely independent of the training paradigm (*Figure 5c*). Hence, the experience-dependence of the model circuit is in line with that of nPE neurons in rodent V1 (*Attinger et al., 2017*).

During learning, we exposed the network to sensory inputs and motor-related predictions designed to reflect coupled sensorimotor experience. To account for changes in the external world that do not arise from the animal's own movements, we included 'playback' phases in which the visual input is stronger than predicted by the motor-related input. Consistent with the experimental setup of *Attinger et al., 2017*, we deliberately excluded feedback mismatch phases. In the model, the stimuli experienced during learning have a strong impact on the response structure of the PCs, because the learning rules aim to keep the PCs at a given baseline rate at all times. The inclusion of feedback and playback phases during learning therefore leads to neurons that remain at their baseline during those phases, in line with nPE neurons. In mouse V1, nPE neurons exhibit an average rate decrease during playback when the animals were only exposed to perfectly coupled sensorimotor experience (*Attinger et al., 2017*). When our network was trained in the same way, we also observed that PCs reduced their firing rate during playback phases (*Figure 5d* and *Figure 5—figure supplement 1*). This can be a result of an excess of somatic inhibition, dendritic inhibition or both. The model hence predicts that the rate reduction during playback phases observed by *Attinger et al., 2017* vanishes when playback phases are included during training.

## nPE circuits can also be learned by biologically plausible learning rules

In our model, nPE neurons developed through inhibitory plasticity that establishes an excitation-inhibition (E/I) balance in PCs. So far, we used learning rules that approximate a backpropagation of error (*Rumelhart et al., 1986*), which changed SOM→PV and VIP→PV connections such as to minimize the difference between the PC firing rate and a baseline rate (see *Equation 16* in Materials and methods). The biological plausibility of such backpropagation rules, which are broadly used in artificial intelligence, is still debated, because they rely on information that is not locally available at the synapse in question (*Crick, 1989*; *Richards and Lillicrap, 2019*). We therefore wondered whether prediction-error circuits can also be established by biologically plausible local learning rules.

We found that nPE neurons also emerged when the backpropagation rules were replaced by a form of plasticity that changes SOM→PV and VIP→PV synapses in proportion to the difference between the excitatory recurrent drive onto PV neurons and a target value (see *Mackwood et al., 2020*, and *Equations 17 and 18* in Materials and methods). This local form of learning was also able

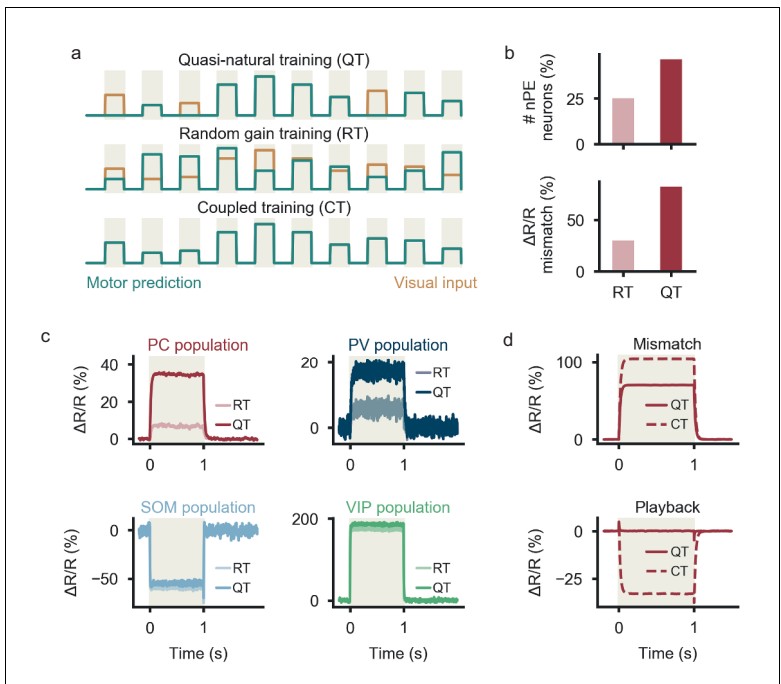

**Figure 5.** Experience-dependence of nPE and PV neurons. (**a**) The network is either exposed to a sequence of baseline, feedback and playback phases (quasi-natural training, QT), to baseline phases and phases during which the visual inputs and motor-related predictions are statistically independent (random gain training, RT) or perfectly coupled sensorimotor experience (coupled training, CT) (**b**) The number of nPE neurons that develop during learning (top) and their mismatch responses (bottom) are smaller for RT than for QT networks. 90% of SOM and 50% of VIP neurons receive visual input. (**c**) Population response ($\Delta R/R$) of PCs, PV, SOM and VIP neurons during mismatch phase. SOM and VIP neurons show the same mismatch response for QT and RT, PCs and PV neurons show stronger responses in QT than in RT. 90% of SOM and 50% of VIP neurons receive visual input. (**d**) Responses during mismatch (top) and playback (bottom) for QT and CT networks. CT networks can exhibit a decrease in activity during playback phase. Connections from VIP to PV neurons are non-plastic and fixed to −0.3. The online version of this article includes the following figure supplement(s) for figure 5:

**Figure supplement 1.** Coupled-trained networks can produce nPE neurons that decrease their activity in playback phase.

to balance excitation and inhibition sufficiently (*Figure 6b,c* and *Figure 1—figure supplement 1c*) so that all PCs developed into nPE neurons (*Figure 6c*).

The plasticity rules can be further simplified when PCs do not receive visual information. In this case, PV neurons also remain at their baseline firing rate in feedback and playback phases (*Figure 2e–f*, right). Hence, the strength of SOM→PV and VIP→PV synapses can be learned according to a homeostatic rule (*Vogels et al., 2011*) that aims to sustain a target rate in the PV neurons (*Figure 6—figure supplement 1* and *Figure 1—figure supplement 1d*, *Equations 19 and 20* in Materials and methods). In summary, the backpropagation-like learning rules for the synapses onto PV neurons can be approximated by biologically plausible rules that exploit local information available at the respective synapses.

## Discussion

How the nervous system disentangles self-generated and external sensory stimuli is a long-standing question (*Bell, 1981*; *Franklin and Wolpert, 2011*; *Keller and Mrsic-Flogel, 2018*). Here, we investigated the circuit level mechanisms that underlie the computation of negative prediction errors and how different types of inhibitory neurons shape these prediction circuits. We used computational modeling to show that nPE neurons can be learned by balancing excitation and inhibition in cortical microcircuits with three types of interneurons. We show that the required E/I balance can be

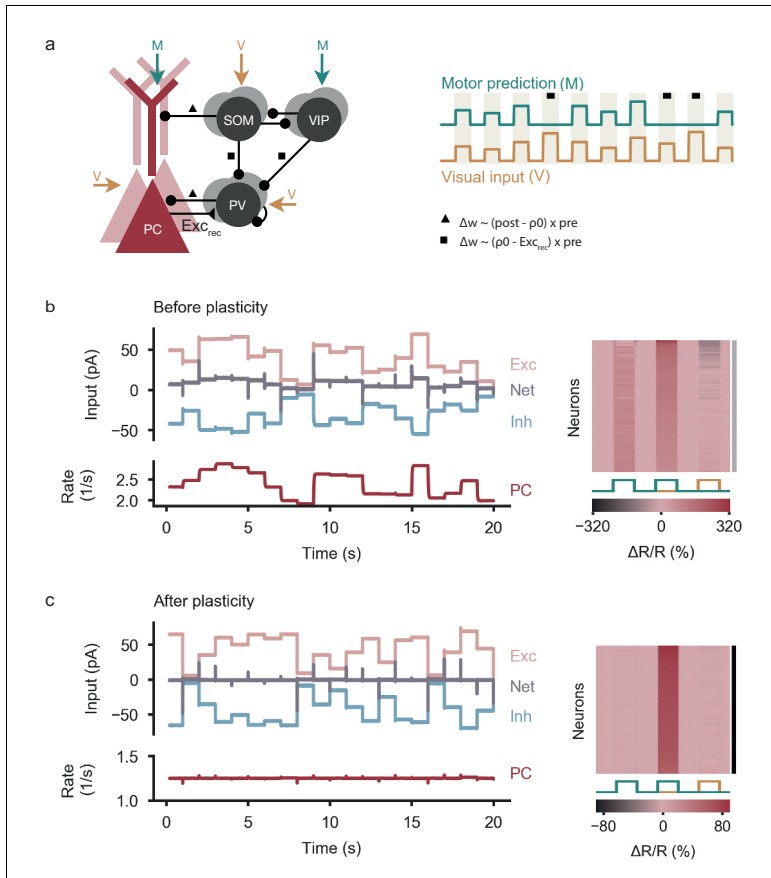

**Figure 6.** Learning nPE neurons by biologically plausible learning rules. (**a**) Left: Network model as in *Figure 1*. Connections marked with symbols undergo experience-dependent plasticity. Connections onto PCs follow an inhibitory plasticity rule akin to *Vogels et al., 2011* (triangle). SOM→PV and VIP→PV synapses change in proportion to the difference between the excitatory recurrent drive onto PV neurons and a target value (square). Right: During plasticity, the network is exposed to a sequence of feedback (coupled sensorimotor experience) and playback phases (black square, visual input not predicted by motor commands). Stimuli last for 1 s and are alternated with baseline phases (absence of visual input and motor predictions). (**b**) Left: Before plasticity, somatic excitation (light red) and inhibition (light blue) in PCs are not balanced. Excitatory and inhibitory currents shifted by ±20 pA for visualization. The varying net excitatory current (gray) causes the PC population rate to deviate from baseline. Right: Response relative to baseline ($\Delta R/R$) of all PCs in feedback, mismatch and playback phases, sorted by amplitude of mismatch response. None of the PCs are classified as nPE neurons (indicated by gray shading to the right). (**c**) Same as in (**b**) after plasticity. Somatic excitation and inhibition are balanced. PC population rate remains at baseline. All PCs classified as nPE neurons (also indicated by black shading to the right).

The online version of this article includes the following figure supplement(s) for figure 6:

**Figure supplement 1.** Learning nPE neurons by biologically plausible learning rules in networks without visual input at the soma of PCs.

achieved by biologically plausible forms of synaptic plasticity. Furthermore, the experience-dependence of the circuit is similar to that of nPE circuits in mouse V1 (*Attinger et al., 2017*).

Our model makes a number of predictions. Firstly, the multi-pathway balance of excitation and inhibition suggests that the input configuration of the prediction circuit could be disambiguated using cell type-specific modulations of neural activity. This could be achieved by optogenetic or pharmacogenetic manipulations, or by exploiting the differential sensitivity of interneuron classes to neuromodulators. The precarious nature of an exact multi-pathway balance also suggests that nPE neurons might change their response characteristics in a context-dependent way, for example by neuromodulatory effects.

Secondly, the central assumption of the model is that nPE neurons emerge by a self-organized E/I balance during sensorimotor experience. It therefore predicts that (i) sensorimotor experience an animal is habituated to should lead to balanced excitation and inhibition in PCs, (ii) E/I balance should break for sensorimotor experience the animal has rarely encountered, for example for mismatches of sensory stimuli and motor predictions and (iii) during altered sensorimotor experience in a virtual reality setting or when the excitability of specific interneuron types is altered, interneuron circuits should gradually reconfigure to reestablish the E/I balance.

PCs in L2/3 of V1 have very low spontaneous firing rates (*Polack et al., 2013*; *Xue et al., 2014*). A potential rate decrease during feedback and playback could hence be hard to detect. Whether the low response of nPE neurons during feedback and playback phases are due to an E/I balance – as suggested here – or due to an excess of inhibition may hence be difficult to decide, and could for example be resolved by intracellular recordings (*Jordan and Keller, 2020*).

We used a mathematical analysis to derive constraints imposed on an interneuron circuit by the presence of nPE neurons. In particular, the equations unveiled the relation between the strength of a number of inhibitory synapses, describing a multi-pathway E/I balance in a network comprising PV, SOM and VIP neurons (see Materials and methods, *Equations 8, 9*). However, we also performed an extensive analysis of different subnetworks, to elucidate under which conditions nPE neurons can emerge (see Appendix 1). By comparing nPE circuits with less cell types, a set of common principles can be extracted (see Appendix 1 for a detailed description): (I) SOM neurons must be present to balance feedback predictions at the dendrites of PCs. (II) SOM neurons must receive visual input unless both PV and VIP neurons are present as well. (III) The connections onto the dendrites must undergo experience-dependent plasticity. (IV) PV neurons must be present when PCs receive visual input in their somatic compartment. (V) Dendritic non-linearities are necessary except for a small set of networks, in which all interneuron types are present and specific constraints for the input configuration apply. While a minimal model that allows nPE neurons to develop comprises SOM neurons and PCs (*Attinger et al., 2017*), the network with three inhibitory neuron types appears the most likely nPE circuit given what is currently known about rodent V1.

The interneuron circuit in our model is motivated by the canonical circuit found in a variety of brain regions (*Pfeffer et al., 2013*; *Lee et al., 2013*; *Jiang et al., 2015*). In addition to the connections between interneuron classes that are frequently reported as strong and numerous, we included VIP→PV synapses in the circuit, because a mathematical analysis reveals that they are required for a perfect E/I balance during both feedback and playback phases (see Appendix 1). While VIP→PV synapses have been found in visual (*Pfeffer et al., 2013*), auditory (*Pi et al., 2013*), somatosensory (*Hioki et al., 2013*; *Lee et al., 2013*) and medial prefrontal cortex (*Pi et al., 2013*), as well as amygdala (*Krabbe et al., 2019*), they are less prominent and often weaker than SOM→PV connections (but see *Krabbe et al., 2019*). VIP→PV synapses can be excluded when the conditions for nPE neurons during feedback and playback phases are mildly relaxed (*Keller and Hahnloser, 2009*; *Keller et al., 2012*; *Attinger et al., 2017*) and when PV neurons receive visual, but not motor inputs (*Figure 1—figure supplement 2*).

Cortical circuits are complex and contain a large variety of interneuron classes (*Rudy et al., 2011*; *Jiang et al., 2015*; *Tremblay et al., 2016*). We restricted the model to three of these classes: PV, SOM and VIP neurons. It is conceivable that several other interneuron types can play a pivotal role in prediction-error circuits. The dendrites of layer 2/3 neurons reach out to layer 1, the major target for feedback connections (*Felleman and Van Essen, 1991*; *Cauller, 1995*; *Larkum, 2013a*) and home to a number of distinct interneuron types (*Larkum, 2013b*; *Schuman et al., 2019*), which may contribute to associative learning (*Abs et al., 2018*). In particular, NDNF neurons unspecifically inhibit apical dendrites located in the superficial layers, and at the same time receive strong inhibition from SOM neurons (*Abs et al., 2018*). Hence, it is possible that these interneurons also shape the processing of feedback information, including the computation of prediction errors.

Our analysis revealed a number of synapses in the circuit that undergo experience-dependent changes. While the synapses from PV neurons onto PCs established a baseline firing rate in the absence of visual input and motor predictions, the synergy between the SOM→PV, VIP→PV and SOM→PC synapses guaranteed that the baseline is retained in feedback and playback phase. The multi-pathway balance of excitation and inhibition could also be achieved by synaptic plasticity in other inhibitory synapses – for example the mutual inhibition between SOM and VIP neurons. However, the assumption that mainly the inhibitory synapses onto PV neurons are plastic is supported by

the observation that PV neuron activity – in contrast to SOM and VIP neuron activity – is experience-dependent (*Attinger et al., 2017*).

In our circuit, the bottom-up and top-down connections conveying actual and predicted visual input, respectively, were non-plastic. However, this modeling choice is not a pre-requisite for the formation of nPE neurons and can be relaxed. As a matter of fact, nPE neurons can also develop in a network, in which the excitatory top-down and bottom-up connections undergo experience-dependent plasticity that balances excitation and inhibition in somatic and dendritic compartments of PCs. For instance, nPE neurons can also be learned by endowing the top-down and bottom-up connections onto PCs and PV neurons with similar plasticity rules described here. Restricting plasticity to the excitatory connections onto PCs would, however, require all inhibitory interneurons to exclusively receive visual input, suggesting that excitatory bottom-up/top-down connections onto interneurons must also change in an activity-dependent manner.

In the model, the plastic inhibitory synapses onto PV neurons change according to non-local information that might not be directly available at the synapse. These synapses therefore implement an approximation of a backpropagation of error, the biological plausibility of which is debated (*Crick, 1989*). We showed that this plasticity rule can be approximated by biologically plausible variants of the plasticity rules. If PCs do not receive direct visual input (*Figure 6—figure supplement 1*), the backpropagation-like algorithm can be replaced by a simple homeostatic Hebbian plasticity rule in the synapses onto the PV interneurons. Given that PCs in V1 are known to receive substantial visual drive (*Yang et al., 2013*; *Xue et al., 2014*), this assumption is unlikely to be valid. We therefore propose an alternative form of plasticity that changes SOM→PV and VIP→PV synapses in proportion to the difference between the excitatory recurrent drive onto PV neurons and a target value (*Mackwood et al., 2020*, see *Figure 6*). The underlying mechanism is similar to feedback alignment (*Lillicrap et al., 2016*) and requires sufficient overlap between the set of postsynaptic PCs a PV neuron inhibits and the set of presynaptic PCs the same PV neuron receives excitation from. This is likely, given the high connection probability between PCs and PV neurons (*Pfeffer et al., 2013*; *Pala and Petersen, 2015*; *Jiang et al., 2015*). Given that the main goal of the present paper was to show that PE circuits can be learned by balancing excitation and inhibition, we used the plasticity rule implementing a backpropagation of error, to ensure maximal generality.

We modeled the apical dendrite of PCs as a single compartment that integrates excitatory and inhibitory input currents and has the potential to produce calcium spike-like events (*Yuste et al., 1994*; *Larkum et al., 1999*; *Murayama et al., 2009*; *Hertäg and Sprekeler, 2019*). Moreover, we assumed that an overshoot of inhibition decouples the apical tuft of the PCs from their soma, by including a rectifying non-linearity that precludes an excess of dendritic inhibition to influence somatic activity. However, the presence or nature of these dendritic nonlinearities has a minor influence on the development of nPE neurons (*Figure 1—figure supplement 3*). When we allowed dendritic inhibition to influence the soma, inhibitory plasticity still established nPE neurons, although the learned interneuron circuit differs with respect to the synaptic strengths. The additional dendritic inhibition reduces the required amount of somatic, PV-mediated inhibition. This is primarily the case during playback phases, when the excitatory motor input to the apical dendrite is absent. PV neurons are therefore less active during the playback phase than during the feedback phase (*Figure 1—figure supplement 3*), consistent with recordings in mouse V1 (*Attinger et al., 2017*).

By modeling the apical dendrite as a single compartment, we also neglected the possibility that dendritic branches process distinct information. However, we expect that the suggested framework of generating predictive signals by a compartment-specific E/I balance generalizes to more complex dendritic configurations, in which local inhibition could contribute by gating different dendritic inputs (*Yang et al., 2016*).

A hallmark of neurons in sensory areas is their pronounced feature selectivity (*Cardin et al., 2007*; *Niell and Stryker, 2008*; *Harris and Mrsic-Flogel, 2013*). This selectivity is also present in nPE neurons in layer 2/3 of rodent V1 which preferentially signal mismatches in a particular location of the visual field (*Zmarz and Keller, 2016*). Here, we did not include feature selectivity, but only modeled one-dimensional input signals representing actual or expected visual input. However, we expect that nPE neurons can also develop in networks in which excitatory neurons are equipped with feature selectivity and receive multi-dimensional inputs, by the same plasticity rules described here. We conjecture that the presence of feature selectivity imposes further constraints on the network, for instance, regarding feature topography or interneuron tuning properties. For future work,

it would be interesting to study how the presence of feature-selective PE neurons constrains the feature selectivity in interneurons that tend to be more broadly tuned than excitatory neurons (*Sohya et al., 2007*; *Cardin et al., 2007*; *Kerlin et al., 2010*; *Atallah et al., 2012*).

In our model, the excitatory recurrent connections target the apical dendrites of PCs, but given that PCs comprise a homogeneous population, they serve no specific computational purpose in the present context. We expect that this would change if the neurons in the circuit were endowed with stimulus selectivity. For instance, the predictive coding model by *Boerlin et al., 2013* assumes separate recurrent loops for coding and computation (see also *Denève and Machens, 2016*). In this model, the membrane potential represents a prediction error and occasional spiking serves the purpose of reducing a potential mismatch by initiating fast inhibition. The excitatory neurons receive feedforward inputs, fast feedback inhibition and slow excitatory recurrent connections. While the fast inhibitory loop balances the excitatory feedforward and the slow feedback inputs, the slower loop – combined with dendritic nonlinearities – allows for nonlinear computations of the delayed represented variable (*Denève and Machens, 2016*). It will be interesting to study how this line of work is related to the PE circuit model we studied here, but it would require to extend the present model to perform richer computations, for example by endowing it with stimulus selectivity.

Here, we have mainly focused on the development of nPE neurons because those have been studied extensively in layer 2/3 of rodent V1, which allowed us to qualitatively compare our model with experimental findings. In contrast, to the best of our knowledge, less is known for pPE neurons in the visual system. Moreover, as we assume that excitatory neurons aim to establish an E/I balance for all stimuli they are regularly exposed to, and as animals experience episodes, in which the change of visual input is only caused by external factors (playback phases), excitatory neurons are more likely to develop into nPE than pPE neurons in the sensorimotor paradigm used here. However, it can be assumed that under different circumstances pPE neurons do play an equally important role in the processing of information. We expect that the same principles and approaches described here also hold for the formation of pPE neurons. Indeed, when a network, in which SOM neurons receive motor-related input and VIP neurons receive visual input, is exposed to baseline, feedback and mismatch phases, pPE neurons develop (see *Appendix 2—figure 2*). The inhibitory plasticity establishes pPE neurons independent of the input configuration onto PCs and PV neurons as long as various excitatory, inhibitory, disinhibitory and dis-disinhibitory pathways can be balanced (see *Appendix 2—figure 1*, *Equations 50 and 51)*.

In the present work, we derived the constraints for separate nPE and pPE neurons and did not study the parallel development of both in the same neural network. While the formation of nPE neurons requires SOM neurons to receive visual input, the formation of pPE neurons requires SOM neurons to receive a motor-related prediction thereof. Given that SOM neurons constitute a heterogeneous population (*Jiang et al., 2015*; *Tremblay et al., 2016*; *Urban-Ciecko and Barth, 2016*), it is conceivable that separate sub-circuits enable the parallel existence of nPE and pPE neurons. However, we expect that the presence of both PE types requires refined constraints on the interneuron circuit and plasticity rules. For instance, the formation of nPE and pPE neurons that only increase their activity in mismatch and playback phases, respectively, while remaining at baseline otherwise, introduces constraints for all three phases. Hence, the network must be exposed to all input phases during learning. In the present framework, this would most likely produce excitatory neurons that remain at their baseline in all phases and hence do not encode prediction errors at all. Hence, the plasticity rules must be modified such that they incorporate gating signals that restrict learning to a subset of input phases or a subset of synapses, for example by controlling the learning rates. It has been argued that specific neuromodulators that are linked to self-motion may guide plasticity in prediction-error circuits (*Keller and Mrsic-Flogel, 2018*). For example, neuromodulators could restrict learning to feedback phases. In this case, excitatory neurons would show deviations from baseline during both playback and mismatch phases, that is essentially all neurons would encode both positive and negative prediction errors. A dichotomy of nPE and pPE neurons could result from low baseline firing rates. A thorough investigation of these scenarios for the simultaneous development of nPE and pPE neurons is, however, beyond the scope of the present study.

Our model suggests a well-orchestrated division of labor of PV, SOM and VIP interneurons that is shaped by experience: While PV neurons balance the sensory input at the somatic compartment of PCs, SOM neurons cancel feedback signals at the apical dendrites. VIP neurons ensure sufficiently large mismatch responses by amplifying small differences between feedforward and feedback inputs

(*Attinger et al., 2017*; *Hertäg and Sprekeler, 2019*). Given the relative uniformity of cortex in its appearance, structure and cell types (*Douglas et al., 1989*; *Mountcastle, 1997*), it is conceivable that the same principles also hold for other regions of the cortex beyond V1. Shedding light on the mechanisms that constitute the predictive power of neuronal circuits may in the long run contribute to an understanding of psychiatric disorders that have long been associated with a malfunction of the brain's prediction machinery (*Fletcher and Frith, 2009*; *Corlett et al., 2009*; *Sinha et al., 2014*; *Lawson et al., 2017*) and specific types of interneurons (*Marín, 2012*; *Hattori et al., 2017*; *Batista-Brito et al., 2018*).

## Materials and methods

### Network model

We simulated a rate-based network model of excitatory pyramidal cells ($N_{PC}$ = 70) and inhibitory PV, SOM and VIP neurons ($N_{PV} = N_{SOM} = N_{VIP}$ = 10). All neurons are randomly connected with connection strengths and probabilities given below (see 'Connectivity').

The excitatory pyramidal cells are described by a two-compartment rate model that was introduced by *Murayama et al., 2009*. The dynamics of the firing rate $r_{E,i}$ of the somatic compartment of neuron $i$ obeys

$$\tau_E \frac{dr_{E,i}}{dt} = -r_{E,i} + [I_i - \Theta], \tag{1}$$

where $\tau_E$ denotes the excitatory rate time constant ($\tau_E$ = 60 ms), $\Theta$ terms the rheobase of the neuron ($\Theta = 14\,s^{-1}$). Firing rates are rectified to ensure positivity. $I_i$ is the total somatic input generated by somatic and dendritic synaptic events and potential dendritic calcium spikes:

$$I_i = \lambda_D \left[ I_{D,i}^{syn} + c_i \right]_+ + (1 - \lambda_E) I_{E,i}^{syn}. \tag{2}$$

Here, the function $[x]_+ = \max(x, 0)$ is a rectifying nonlinearity that prohibits an excess of inhibition at the apical dendrite to reach the soma. $I_{D,i}^{syn}$ and $I_{E,i}^{syn}$ are the total synaptic inputs into dendrite and soma, respectively, and $c_i$ denotes a dendritic calcium event. $\lambda_D$ and $\lambda_E$ are the fractions of 'currents' leaking away from dendrites and soma, respectively ($\lambda_D$=0.27, $\lambda_E$=0.31). The synaptic input to the soma $I_{E,i}^{syn}$ is given by the sum of external sensory inputs $x_E$ and PV neuron-induced (P) inhibition,

$$I_{E,i}^{syn} = x_E - \sum_{j=1}^{N_{PV}} w_{EP,ij} \cdot r_{P,j}. \tag{3}$$

The dendritic input $I_{D,i}^{syn}$ is the sum of motor-related predictions $x_D$, the recurrent, excitatory connections from other PCs and SOM neuron-induced (S) inhibition:

$$I_{D,i}^{syn} = x_D - \sum_{j=1}^{N_{SOM}} w_{DS,ij} \cdot r_{S,j} + \sum_{j=1}^{N_{PC}} w_{DE,ij} \cdot r_{E,j}. \tag{4}$$

The weight matrices $w_{EP}$, $w_{DS}$ and $w_{DE}$ denote the strength of connection between PV neurons and the soma of PCs ($w_{EP}$), SOM neurons and the dendrites of PCs ($w_{DS}$) and the recurrence between PCs ($w_{DE}$), respectively. The input generated by a calcium spike is given by

$$c_i = c \cdot H(I_{D,i}^0 - \Theta_c), \tag{5}$$

where $c$ scales the amount of current produced ($c = 7\,s^{-1}$), $H$ is the Heaviside step function, $\Theta_c$ represents a threshold that describes the minimal input needed to produce a $Ca^{2+}$-spike ($\Theta_c = 28\,s^{-1}$) and $I_{D,i}^0$ denotes the total, synaptically generated input in the dendrites,

$$I_{D,i}^0 = \lambda_E I_{E,i}^{syn} + (1 - \lambda_D) I_{D,i}^{syn}. \tag{6}$$

Note that we incorporated the gain factor present in *Murayama et al., 2009* into the parameters

to achieve unit consistency for all neuron types (when we compared excitatory/inhibitory currents, the respective activities were divided by this gain factor, $g = 0.07$ (pA· s)$^{-1}$).

The firing rate dynamics of each interneuron is modeled by a rectified, linear differential equation (*Wilson and Cowan, 1972*),

$$\tau_i \frac{dr_{X,i}}{dt} = -r_{X,i} + \sum_{j=1}^{N_{PC}} w_{XE,ij} \cdot r_{E,j} - \sum_{j=1}^{N_{PV}} w_{XP,ij} \cdot r_{P,j} - \sum_{j=1}^{N_{SOM}} w_{XS,ij} \cdot r_{S,j} - \sum_{j=1}^{N_{VIP}} w_{XV,ij} \cdot r_{V,j} + x_i, \tag{7}$$

where $r_{X,i}$ denotes the firing rate of neuron $i$ from neuron type $X$ ($X \in \{P,S,V\}$) and $x_i$ represents external inputs. The weight matrices $w_{XY}$ denote the strength of connection between the presynaptic neuron population $Y$ and the postsynaptic neuron population $X$. The rate time constant $\tau_i$ was chosen to resemble a fast GABA$_A$ time constant, and set to 2 ms for all interneuron types included.

## Negative prediction-error neurons

We define PCs as nPE neurons when they exclusively increase their firing rate during feedback mismatch (visual input smaller than predicted), while remaining at their baseline during feedback and playback phases. In a linearized, homogeneous network and under the assumption that the apical dendrites are sufficiently inhibited during feedback and playback phase, this definition is equivalent to two constraints on the interneuron network (see Appendix 1 for a detailed analysis and derivation):

$$w_{PS} = V_P + w_{VS} M_P - \frac{(1+w_{PP})}{w_{EP}} V_E, \tag{8}$$

$$w_{PV} = M_P + w_{SV} V_P - w_{SV} \frac{(1+w_{PP})}{w_{EP}} V_E$$

$$= w_{SV} w_{PS} + (1 - w_{SV} w_{VS}) M_P. \tag{9}$$

The parameters $V_X, M_X \in \{0,1\}$ indicate whether neuron type $X$ receives visual and motor-related inputs, respectively, and control the different input configurations. In addition to the conditions *Equations 8 and 9*, the synapses from SOM neurons onto the apical dendrites must be sufficiently strong to cancel potential excitatory inputs during feedback and playback phase.

In practice, we classify PCs as nPE neurons when $\Delta R/R$ is larger than 20% in the mismatch phase and less than ±10% elsewhere ($\Delta R/R = (r - r_{BL})/r_{BL}$, $r_{BL}$: baseline firing rate). Tolerating small deviations in feedback and playback phase is more in line with experimental approaches. The results do not rely on the precise thresholds used for the classification.

## Connectivity

All neurons are randomly connected with connection probabilities motivated by the experimental literature (*Fino and Yuste, 2011*; *Packer and Yuste, 2011*; *Pfeffer et al., 2013*; *Lee et al., 2013*; *Pi et al., 2013*; *Jiang et al., 2015*; *Jouhanneau et al., 2015*; *Pala and Petersen, 2015*),

$$p = \begin{pmatrix} p_{EE} & p_{EP} & p_{ES} & p_{EV} \\ p_{DE} & p_{DP} & p_{DS} & p_{DV} \\ p_{PE} & p_{PP} & p_{PS} & p_{PV} \\ p_{SE} & p_{SP} & p_{SS} & p_{SV} \\ p_{VE} & p_{VP} & p_{VS} & p_{VV} \end{pmatrix} = \begin{pmatrix} - & 0.6 & - & - \\ 0.1 & - & 0.55 & - \\ 0.45 & 0.5 & 0.6 & 0.5 \\ 0.35 & - & - & 0.5 \\ 0.1 & - & 0.45 & - \end{pmatrix}. \tag{10}$$

All cells of the same neuron type have the same number of incoming connections. The mean connection strengths are given by

$$w = \begin{pmatrix} w_{EE} & w_{EP} & w_{ES} & w_{EV} \\ w_{DE} & w_{DP} & w_{DS} & w_{DV} \\ w_{PE} & w_{PP} & w_{PS} & w_{PV} \\ w_{SE} & w_{SP} & w_{SS} & w_{SV} \\ w_{VE} & w_{VP} & w_{VS} & w_{VV} \end{pmatrix} = \begin{pmatrix} - & * & - & - \\ 0.42 & - & * & - \\ * & * & * & * \\ 1 & - & - & 0.6 \\ 1 & - & 0.5 & - \end{pmatrix} \tag{11}$$

where the symbol * denotes weights that vary between simulations (e.g., subject to plasticity or

computed from the *Equations 8 and 9*). For non-plastic networks, these synaptic strengths are given by $w_{\text{EP}} = 2.8$, $w_{\text{DS}} = 3.5$, $w_{\text{PE}} = 1.5$, $w_{\text{PP}} = 0.1$ (if PCs receive visual input) or $w_{\text{PP}} = 1.5$ (if PCs receive no visual input), $w_{\text{PS}}$ and $w_{\text{PV}}$ are computed from the *Equations 8 and 9*.

For plastic networks, the initial connections between neurons are drawn from uniform distributions $w_{ij}^{initial} \in \mathcal{U}(0.5\ w, 1.5\ w)$ where $w$ denotes the mean connection strengths given in (*Equation 11*) and $w_{\text{EP}} = 1.75$, $w_{\text{DS}} = 0.35$, $w_{\text{PE}} = 2.5$ (if PCs receive visual input) or $w_{\text{PE}} = 1.2$ (if PCs receive no visual input), $w_{\text{PP}} = 0.5$ (if PCs receive visual input) or $w_{\text{PP}} = 1.5$ (if PCs receive no visual input), $w_{\text{PS}} = 0.3$ and $w_{\text{PV}} = 0.6$. Please note that the system is robust to the choice of connections strengths. The connection strengths are merely chosen such that the solutions of *Equations 8 and 9* comply with Dale's principle.

All weights are scaled in proportion to the number of existing connections (i.e., the product of the number of presynaptic neurons and the connection probability), so that the results are independent of the population size.

## Inputs

All neurons receive constant, external background input that ensures reasonable baseline firing rates in the absence of visual and motor-related input. In the case of non-plastic networks, these inputs were set such that the baseline firing rates are $r_{\text{E}} = 1s^{-1}$, $r_{\text{P}} = 2s^{-1}$, $r_{\text{S}} = 2s^{-1}$ and $r_{\text{V}} = 4s^{-1}$. In the case of plastic networks, we set the external inputs to $x_{\text{E}} = 28s^{-1}$, $x_{\text{D}} = 0s^{-1}$, $x_{\text{P}} = 2s^{-1}$, $x_{\text{S}} = 2s^{-1}$ and $x_{\text{V}} = 2s^{-1}$ (if not stated otherwise). In addition to the external background inputs, the neurons receive either visual input ($v$), a motor-related prediction thereof ($m$) or both.

In line with the experimental setup of *Attinger et al., 2017*, we distinguish between baseline ($m = v = 0$), feedback ($m = v > 0$), feedback mismatch ($m > v$) and playback ($m < v$) phases. During training, the network is exposed to feedback and playback phases with stimuli drawn from a uniform distribution from the interval $[0, 7s^{-1}]$. After learning, the strength of stimuli is set to $7s^{-1}$ (plastic networks) or $3.5s^{-1}$ (non-plastic networks).

## Plasticity

In plastic networks, a number of connections between neurons are subject to experience-dependent changes in order to establish an E/I balance for PCs. PV→PC and the PC→PV synapses establish the target firing rates for PCs and PV neurons, respectively. VIP→PV and SOM→PV synapses and the synapses from SOM neurons onto the apical dendrites of PCs ensure that PCs remain at their baseline during feedback and playback phase. The corresponding plasticity rules are of the form

$$\Delta w \propto \pm (\text{post} - \text{baseline}) \cdot \text{pre} \tag{12}$$

### Connections onto PCs

In detail, the connections from PV and SOM neurons onto the soma and the apical dendrites, respectively, obey inhibitory Hebbian plasticity rules akin to *Vogels et al., 2011*

$$\Delta w_{\text{EP},ij} \propto (r_{\text{E},i}^{post} - \rho_{\text{E},0}^{post}) \cdot r_{\text{P},j}^{pre}, \tag{13}$$

$$\Delta w_{\text{DS},ij} \propto (A_i^{post} - \epsilon) \cdot r_{\text{S},j}^{pre}. \tag{14}$$

The parameter $\rho_{\text{E},0}^{post}$ denotes the baseline firing rate of the postsynaptic PC, and the dendritic activity $A_i^{post}$ is given by the rectified synaptic events at the dendrites

$$A_i^{post} = \left[ I_{\text{D},i}^{\text{syn}} + c_i \right]_+ . \tag{15}$$

The small 'correction' term $\epsilon$ eases the effect of strong onset responses (here, we used $\epsilon = 0.1s^{-1}$).

### Connections onto PV neurons - non-local learning

The connections from both SOM and VIP neurons onto PV neurons implement an approximation of a backpropagation of error

$$\Delta w_{ij} \propto \frac{1}{N_{\mathrm{E},i}} \sum_{k \in S_i^{post}} (\rho_{\mathrm{E},0}^{post} - r_{\mathrm{E},k}^{post}) \cdot r_j^{pre}. \tag{16}$$

$S_i^{post}$ denotes the set of postsynaptic PCs a particular PV neuron is connected to, and $N_{\mathrm{E},i}$ is the number of excitatory neurons in $S_i^{post}$.

## Connections onto PV neurons - local approximation to backpropagation of error

When the connection probability between PCs and PV neurons is large, this backpropagation of error can be replaced by a biologically plausible learning rule that only relies on local information available in the PV neurons (*Figure 6*),

$$\Delta w_{ij} \propto \Delta \mathrm{E}_{\mathrm{rec},i} \cdot r_j^{pre}, \tag{17}$$

where $\Delta \mathrm{E}_{\mathrm{rec},i}$ denotes the difference between the excitatory recurrent drive onto PV neuron $i$ and a target value

$$\Delta \mathrm{E}_{\mathrm{rec},i} = \sum_{k \in S_i^{pre}} w_{\mathrm{PE},ik} \cdot (\rho_{\mathrm{E},0}^{post} - r_{\mathrm{E},k}^{post}). \tag{18}$$

$S_i^{pre}$ denotes the set of presynaptic PCs a particular PV neuron receives excitation from.

## Connections onto PV neurons - learning with a homeostatic firing rate for PV neurons

When nPE neurons do not receive direct visual input, the backpropagation rules can be simplified even further (*Figure 6—figure supplement 1*). The synapses onto PV neurons can be learned according to a Hebbian inhibitory plasticity rule (*Vogels et al., 2011*) that aims to sustain a baseline rate in the PV neurons

$$\Delta w_{\mathrm{PX},ij} \propto (r_{\mathrm{P},i}^{post} - \rho_{\mathrm{P},0}^{post}) \cdot r_{\mathrm{X},j}^{pre} \tag{19}$$

with $X \in \{S, V\}$. This baseline rate is established by modifying the connections from PCs onto PV neurons according to an anti-Hebbian plasticity rule

$$\Delta w_{\mathrm{PE},ij} \propto (\rho_{\mathrm{P},0}^{post} - r_{\mathrm{P},i}^{post}) \cdot r_{\mathrm{E},j}^{pre}. \tag{20}$$

### Simulation and code availability

All simulations were performed in customized Python code written by LH. Differential equations were numerically integrated using a 2nd-order Runge-Kutta method with time steps between 0.05 and 2 ms. Neurons were initialized with $r_i(0) = 0$. Source code and data for all figures will be available after publication at *Hertäg, 2020* (https://github.com/sprekelerlab/SourceCode_Hertaeg20, copy archived at https://github.com/elifesciences-publications/SourceCode_Hertaeg20).

### Acknowledgements

We are grateful to Laura Bella Naumann and Joram Keijser for critical reading of the manuscript and Owen Mackwood for technical guidance during development of the simulator. We also want to thank all members of the Sprekeler lab for discussion, support and comments on the manuscript. The project is funded by the German Federal Ministry for Education and Research, FKZ 01GQ1201 and the DFG via the collaborative research center FOR 2143.

## Additional information

### Funding

| Funder | Grant reference number | Author |
|---|---|---|
| Bundesministerium für Bildung und Forschung | FKZ 01GQ1201 | Henning Sprekeler |
| Deutsche Forschungsgemeinschaft | FOR 2143 | Henning Sprekeler |

The funders had no role in study design, data collection and interpretation, or the decision to submit the work for publication.

### Author contributions

Loreen Hertäg, Conceptualization, Data curation, Software, Formal analysis, Validation, Investigation, Visualization, Methodology, Writing - original draft, Writing - review and editing; Henning Sprekeler, Conceptualization, Resources, Supervision, Funding acquisition, Writing - original draft, Project administration, Writing - review and editing

### Author ORCIDs

Loreen Hertäg (iD) https://orcid.org/0000-0001-7838-3361
Henning Sprekeler (iD) http://orcid.org/0000-0003-0690-3553

### Decision letter and Author response

Decision letter https://doi.org/10.7554/eLife.57541.sa1
Author response https://doi.org/10.7554/eLife.57541.sa2

## Additional files

### Supplementary files

• Transparent reporting form

### Data availability

Source code to reproduce simulated data and figures is publicly available at https://github.com/sprekelerlab/SourceCode_Hertaeg20 (copy archived at https://github.com/elifesciences-publications/SourceCode_Hertaeg20).

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

# Appendix 1

We performed a mathematical analysis of a simplified model to identify the constraints that are imposed on the interneuron circuit by the presence of nPE neurons. We first describe the assumptions made and the definition of nPE neurons. We then derive the constraints for a simplified network with canonical interneuron connectivity including VIP-to-PV synapses. The solutions provide the relationship for the strength of synapses between different neuron types that must be satisfied for nPE neurons to emerge. We then show that the same network without VIP-to-PV synapses can only produce nPE neurons under very restrictive assumptions. Finally, we will provide an exhaustive list of networks, which differ in terms of number of interneuron types, their inputs and the nature of the dendritic processes, and specify the conditions under which these networks can produce nPE neurons.

## Constraints for the interneuron circuit by the presence of nPE neurons

To derive the constraints for the interneuron network that are imposed by the presence of nPE neurons, we performed a mathematical analysis of a simplified network model, in which the nonlinearity of the dendritic compartment and the rectifying nonlinearities are neglected. This reduces the network to an analytically tractable linear system. The simplifications rely on the following assumptions:

1. During baseline, feedback and playback phases, SOM interneuron-mediated inhibition either equals or exceeds excitatory motor predictions arriving at the apical dendrites of PCs.
2. Any excess of inhibition in the dendrite does not affect the soma of PCs.
3. During baseline, feedback and playback phases, all neuron types have positive firing rates, such that the rate rectification can be neglected.

These assumptions allow us to omit the dendritic compartment of PCs and consequently all synapses thereto. The remaining system of linear equations describes the activity of all neuron types during baseline, feedback and playback phase. For the subsequent analysis, we furthermore consider a homogeneous network, that is, all weights, neuronal properties and the number of incoming connections for cells of the same type are the same. As a result, we can reduce the high-dimensional system to four equations, each describing the dynamics of one representative firing rate per neuron type:

$$\tau \frac{d\mathbf{r}}{dt} = -\mathbf{r} + \Omega\mathbf{r} + \mathbf{X}, \tag{21}$$

where $\tau$ denotes the rate time constant, $\mathbf{r} = [r_\mathrm{E}, r_\mathrm{P}, r_\mathrm{S}, r_\mathrm{V}]^T$ (subscripts refer to the different neuron types; E: soma of PC, P: PV, S: SOM, V: VIP), $\Omega$ is the weight matrix and $\mathbf{X}$ denotes the external inputs. In the steady state, the firing rates are given by

$$\mathbf{r} = -(\Omega - \mathbb{1})^{-1}\mathbf{X} = W^{-1}\mathbf{X} \tag{22}$$

with the effective connectivity matrix $W$ that includes the leak:

$$W = \begin{pmatrix} -1 & -w_\mathrm{EP} & 0 & 0 \\ w_\mathrm{PE} & -1-w_\mathrm{PP} & -w_\mathrm{PS} & -w_\mathrm{PV} \\ w_\mathrm{SE} & -w_\mathrm{SP} & -1-w_\mathrm{SS} & -w_\mathrm{SV} \\ w_\mathrm{VE} & -w_\mathrm{VP} & -w_\mathrm{VS} & -1-w_\mathrm{VV} \end{pmatrix}. \tag{23}$$

The weight parameters $w_{XY}$ between neuron types are strictly positive to maintain the excitatory/inhibitory nature of the various neuron types. In our model, an excitatory neuron is classified as a perfect nPE neuron, if

$$r_\mathrm{E}^{(feedback)} = r_\mathrm{E}^{(playback)} = r_\mathrm{E}^{(baseline)}, \tag{24}$$

$$r_\mathrm{E}^{(mismatch)} > r_\mathrm{E}^{(baseline)}. \tag{25}$$

During feedback mismatch, the PC firing rate increases with respect to the baseline as long as the motor-related excitatory inputs exceed the somatic inhibition mediated by PV neurons. The

conditions according to which no change in activity occurs in either feedback or playback phase (see *Equation 24*) impose constraints on the weight configuration that need to be satisfied. These can be summarized by

$$0 = W^{-1}\mathbf{X}^{fb}, \tag{26}$$

$$0 = W^{-1}\mathbf{X}^{pb}, \tag{27}$$

where $\mathbf{X}^{fb}$ and $\mathbf{X}^{pb}$ denote the excess external inputs above baseline during feedback and playback phase, respectively,

$$\mathbf{X}^{fb} = [V_{\mathrm{E}}, V_{\mathrm{P}} + M_{\mathrm{P}}, 1, 1]^T \cdot s, \tag{28}$$

$$\mathbf{X}^{pb} = [V_{\mathrm{E}}, V_{\mathrm{P}}, 1, 0]^T \cdot s, \tag{29}$$

with $s$ representing a varying excitatory stimulus strength. The parameters $V_X, M_X \in \{0, 1\}$ indicate whether neuron type $X$ receives visual and motor-related inputs, respectively, and control the different input configurations.

## Interneuron connectivity with VIP-to-PV synapses

We start with the connectivity motif proposed by *Pfeffer et al., 2013*. We also allow for connections from VIP to PV neurons. Although they are considered to be less prominent and weaker than connections from VIP to SOM neurons and are therefore often neglected in diagrams and computational models, those synapses have been observed in various brain regions (*Pi et al., 2013*; *Pfeffer et al., 2013*; *Hioki et al., 2013*; *Lee et al., 2013*; *Krabbe et al., 2019*). To this end, the respective connectivity matrix is given by

$$W = \begin{pmatrix} -1 & -w_{\mathrm{EP}} & 0 & 0 \\ w_{\mathrm{PE}} & -1 - w_{\mathrm{PP}} & -w_{\mathrm{PS}} & -w_{\mathrm{PV}} \\ w_{\mathrm{SE}} & 0 & -1 & -w_{\mathrm{SV}} \\ w_{\mathrm{VE}} & 0 & -w_{\mathrm{VS}} & -1 \end{pmatrix}. \tag{30}$$

The constraints (*Equation 26*) and (*Equation 27*) defining nPE neurons are then given by

$$0 = (1 - w_{\mathrm{SV}}w_{\mathrm{VS}})(1 + w_{\mathrm{PP}})V_{\mathrm{E}} - w_{\mathrm{EP}}(1 - w_{\mathrm{SV}}w_{\mathrm{VS}})(V_{\mathrm{P}} + M_{\mathrm{P}})$$
$$+ w_{\mathrm{EP}}w_{\mathrm{PS}}(1 - w_{\mathrm{SV}}) + w_{\mathrm{EP}}w_{\mathrm{PV}}(1 - w_{\mathrm{VS}}), \tag{31}$$
$$0 = (1 - w_{\mathrm{SV}}w_{\mathrm{VS}})(1 + w_{\mathrm{PP}})V_{\mathrm{E}} - w_{\mathrm{EP}}(1 - w_{\mathrm{SV}}w_{\mathrm{VS}})V_{\mathrm{P}} + w_{\mathrm{EP}}(w_{\mathrm{PS}} - w_{\mathrm{PV}}w_{\mathrm{VS}}). \tag{32}$$

These two equations yield

$$w_{\mathrm{PS}} = V_{\mathrm{P}} + w_{\mathrm{VS}}M_{\mathrm{P}} - \frac{(1 + w_{\mathrm{PP}})}{w_{\mathrm{EP}}}V_{\mathrm{E}}, \tag{33}$$

$$w_{\mathrm{PV}} = M_{\mathrm{P}} + w_{\mathrm{SV}}\, V_{\mathrm{P}} - w_{\mathrm{SV}}\frac{(1 + w_{\mathrm{PP}})}{w_{\mathrm{EP}}}V_{\mathrm{E}}$$
$$= w_{\mathrm{SV}}w_{\mathrm{PS}} + (1 - w_{\mathrm{SV}}w_{\mathrm{VS}})M_{\mathrm{P}}. \tag{34}$$

*Equation 33 and 34* are the mathematical formulation of the E/I balance of multiple pathways shown in *Figure 2* and *Figure 2—figure supplement 1*.

For the derivation above, we have assumed that the motor-related input is switched off during the playback phase. This assumption, however, can be relaxed. When motor predictions are merely smaller than the actual sensory input but non-zero during playback, analogous calculations yield the same constraints.

## Interneuron connectivity without VIP-to-PV synapses

Without connections from VIP onto PV neurons, the constraints (*Equation 26*) and (*Equation 27*) yield

$$0 = (1 - w_{SV}w_{VS})(1 + w_{PP})V_E - w_{EP}(1 - w_{SV}w_{VS})(V_P + M_P) + w_{EP}w_{PS}(1 - w_{SV}), \quad (35)$$

$$0 = (1 - w_{SV}w_{VS})(1 + w_{PP})V_E - w_{EP}(1 - w_{SV}w_{VS})V_P + w_{EP}w_{PS}. \quad (36)$$

These two equations simplify to

$$w_{PS} = \frac{(w_{SV}w_{VS} - 1)}{w_{SV}}M_P. \quad (37)$$

As the weight $w_{PS}$ is strictly positive (see definition of weight matrix above), the product $w_{SV}w_{VS}$ must be larger than 1. This, however, indicates that networks with rate rectification exceed a bifurcation point and run into a winner-take-all (WTA) regime, in which either VIP or SOM neurons are silent (*Hertäg and Sprekeler, 2019*).

With VIP neurons being silent in all phases but during feedback mismatch phases, the constraint on $w_{PS}$ can be recalculated from *Equations 40 and 24* while neglecting VIP neurons:

$$w_{PS} = V_P - \frac{(1 + w_{PP})}{w_{EP}}V_E. \quad (38)$$

This equation reveals that PV neurons must receive visual input to ensure $w_{PS} > 0$.

In summary, this mathematical analysis shows that perfect nPE neurons can only emerge when VIP neurons are silent during all phases but the feedback mismatch phase.

Please note that the very same results are obtained even if connections from PV to both SOM and VIP neurons are included.

## Summary of nPE circuits and their constraints

To derive the weight and input constraints on nPE circuits for networks with different complexity, we varied the number of interneuron types and the nature of the dendritic processes. The following network features were not varied, either because they are strongly constrained by what is known about the circuit or because the number of variations would become too large:

- We require nPE neurons to remain at baseline during feedback and playback phases (perfect nPE neurons).
- We consider a canonical microcircuit in which both VIP and SOM neurons inhibit other interneuron types but not themselves and SOM neurons inhibit the apical dendrites of PCs, while PV neurons inhibit the soma of PCs and other PV neurons.
- PCs receive a motor-related prediction of visual input at their dendrites.
- All interneurons receive either visual or motor-related input.
- During baseline, feedback and playback phases all neuron types have non-zero firing rates.
- During feedback, motor-related inputs and the excitatory recurrence at the dendrite are perfectly balanced by SOM neuron mediated inhibition.

In contrast to the assumption made before according to which the dendrite is rectified, we now also consider the possibility that an excess of inhibition can be forwarded to the soma. Following the same approaches outlined in the sections above, and taking into account the assumptions herein before mentioned, we derive a few unifying principles that are required for nPE neurons to develop:

- SOM interneurons must be present to provide dendritic inhibition
- The synapses from SOM neurons onto the dendrites of PCs must undergo experience-dependent plasticity to achieve an E/I balance at the dendrites during feedback.
- When PCs receive visual input, PV neurons must be present (to provide somatic inhibition that can balance the inputs at the soma).
- Only when PV, SOM and VIP neurons are present, dendritic non-linearities may not be strictly necessary
- SOM neurons must receive visual input unless PV and VIP neurons are present.

This leaves four networks with a number of constraints, detailed below.

### PC-SOM

In a network comprising PCs and SOM neurons, PCs act as nPE neurons when

- the dendrites are rectified,
- SOM neurons receive visual input, while PCs do not, and
- $w_{\mathrm{DS}}$ is tuned (see *Equations 4 and 7*)

$$w_{\mathrm{DS}} = \frac{x_{\mathrm{D}} + s_{\max} + w_{\mathrm{DE}}\rho_E}{x_{\mathrm{S}} + s_{\max} + w_{\mathrm{SE}}\rho_E} \to 1 \text{ for } s_{\max} \to \infty$$

where $s_{\max}$ denotes the maximal stimulus strength and $\rho_E$ represents the PC baseline firing rate.

Mismatch responses are caused by an excess of dendritic excitation that cannot be canceled by SOM neuron mediated inhibition.

## PC-SOM-VIP

In a network comprising PCs and SOM and VIP neurons, PCs act as nPE neurons when

- the dendrites are rectified,
- SOM neurons receive visual input, while PCs do not, and
- $w_{\mathrm{DS}}$ is tuned (see *Equations 4 and 7*)

$$w_{\mathrm{DS}} = \frac{(x_{\mathrm{D}} + s_{\max} + w_{\mathrm{DE}}\rho_E) \cdot (1 - w_{\mathrm{SV}}w_{\mathrm{VS}})}{x_{\mathrm{S}} - w_{\mathrm{SV}}x_{\mathrm{V}} + s_{\max}(1 - w_{\mathrm{SV}}) + (w_{\mathrm{SE}} - w_{\mathrm{SV}}w_{\mathrm{VE}})\rho_E} \to \frac{1 - w_{\mathrm{SV}}w_{\mathrm{VS}}}{1 - w_{\mathrm{SV}}} \text{ for } s_{\max} \to \infty$$

where $s_{\max}$ denotes the maximal stimulus strength and $\rho_E$ represents the PC baseline firing rate.

Mismatch responses are caused by an excess of dendritic excitation that cannot be canceled by SOM neuron mediated inhibition.

## PC-PV-SOM

In a network comprising PCs and SOM and PV neurons, PCs act as nPE neurons when

- SOM neurons receive visual input,
- $w_{\mathrm{DS}}$ is tuned (see *Equations 4 and 7*)

$$w_{\mathrm{DS}} = \frac{x_{\mathrm{D}} + s_{\max} + w_{\mathrm{DE}}\rho_E}{x_{\mathrm{S}} + s_{\max} + w_{\mathrm{SE}}\rho_E} \to 1 \text{ for } s_{\max} \to \infty$$

where $s_{\max}$ denotes the maximal stimulus strength and $\rho_E$ represents the PC baseline firing rate,

- somatic excitation and disinhibition must be balanced by PV mediated inhibition (see *Equation 8*)

$$w_{\mathrm{PS}} = 1 - \frac{(1 + w_{\mathrm{PP}})}{w_{\mathrm{EP}}}V_E$$

- and PV neurons must receive visual input (if the dendrites are rectified) or motor predictions thereof (if the dendrites are not rectified).

Mismatch responses are either caused by an excess of dendritic excitation that cannot be canceled by SOM neuron mediated inhibition (if the dendrites are rectified) or rely on the presence of supra-linear processes in the dendrites – for instance calcium spikes (if the dendrites are not rectified).

## PC-PV-SOM-VIP

In a network comprising PCs and SOM, VIP and PV neurons, PCs act as nPE neurons when the following conditions are met:

- $w_{DS}$ is tuned (see **Equations 4 and 7**)

$$w_{DS} = \frac{(x_D + s_{max} + w_{DE}\rho_E) \cdot (1 - w_{SV}w_{VS})}{x_S - w_{SV}x_V + s_{max}(1 - w_{SV}) + (w_{SE} - w_{SV}w_{VE})\rho_E} \rightarrow \frac{1 - w_{SV}w_{VS}}{1 - w_{SV}}$$

where $s_{max}$ denotes the maximal stimulus strength and $\rho_E$ represents the PC baseline firing rate.

- Dependent on the input onto SOM/VIP neurons and the nature of the dendritic processes:
  - If both SOM and VIP neurons receive visual input and dendrites are rectified, then (see **Equation 31**)

$$\begin{aligned} V_P &= 1, \\ w_{PS}(1 - w_{SV}) + w_{PV}(1 - w_{VS}) &= (1 - w_{SV}w_{VS})\left(1 - \frac{(1+w_{PP})}{w_{EP}}V_E\right). \end{aligned}$$

    Mismatch responses are caused by an excess of dendritic excitation that cannot be canceled by SOM neuron mediated inhibition.
  - If both SOM and VIP neurons receive visual input and dendrites are not rectified, then (see **Equation 31** and **Equations 2–4 and 7**)

$$\begin{aligned} M_P &= 1, \\ w_{PS}(1 - w_{SV}) + w_{PV}(1 - w_{VS}) &= (1 - w_{SV}w_{VS})\left(1 - \frac{(1+w_{PP})}{w_{EP}}V_E\right), \\ w_{EP} &= \frac{\lambda_D}{(1-\lambda_E)} \cdot (1 + w_{PP}). \end{aligned}$$

    Mismatch responses are caused by an excess of dendritic excitation that cannot be canceled by SOM neuron mediated inhibition and over-compensates for increased somatic inhibition.
  - If SOM neurons receive visual input, while VIP neurons receive a motor-related prediction thereof and dendrites are rectified, then (see **Equations 8 and 9**)

$$\begin{aligned} w_{PS} &= V_P + w_{VS}M_P - \frac{(1+w_{PP})}{w_{EP}}V_E, \\ w_{PV} &= M_P + w_{SV}V_P - w_{SV}\frac{(1+w_{PP})}{w_{EP}}V_E. \end{aligned}$$

    Mismatch responses are caused by an excess of dendritic excitation that cannot be canceled by SOM neuron mediated inhibition.
  - If SOM neurons receive visual input, while VIP neurons receive a motor-related prediction thereof and dendrites are not rectified, then (see **Equations 8 and 9**, augmented with an inhibitory dendritic current $\Phi_{ED}$ that influences the soma during playback)

$$\begin{aligned} w_{PS} &= V_P + w_{VS}M_P - \frac{(1+w_{PP})}{w_{EP}}V_E + \frac{(1-w_{VS})\cdot\Phi_{ED}}{(1-w_{SV}w_{VS})\cdot w_{EP}}, \\ w_{PV} &= M_P + w_{SV}V_P - w_{SV}\frac{(1+w_{PP})}{w_{EP}}V_E - \frac{\Phi_{ED}}{w_{EP}}, \end{aligned}$$

    where $\Phi_{ED}$ is given by (**Equations 2–4 and 7**)

$$\Phi_{ED} = \frac{\lambda_D}{(1-\lambda_E)} \cdot \frac{(1 - w_{SV}w_{VS})}{(1 - w_{SV})} \cdot (1 + w_{PP}).$$

    Mismatch responses are caused by an excess of dendritic excitation that cannot be canceled by SOM neuron mediated inhibition and over-compensates for increased somatic inhibition.

- If SOM neurons receive motor input, while VIP neurons receive visual input (see *Equations 8 and 9*, augmented with an excitatory dendritic current $\Phi_{ED}$ that influences the soma during playback)

$$
\begin{aligned}
w_{PV} &= V_P + w_{SV}\, M_P - \frac{(1+w_{PP})}{w_{EP}} V_E - \frac{(1-w_{SV})\cdot \Phi_{ED}}{(1-w_{SV}w_{VS})\cdot w_{EP}}, \\
w_{PS} &= w_{VS}\, w_{PV} + (1 - w_{SV}w_{VS})\, M_P + \frac{\Phi_{ED}}{w_{EP}},
\end{aligned}
$$

where $\Phi_{ED}$ is given by (*Equations 2–4 and 7*)

$$
\Phi_{ED} = \frac{\lambda_D}{(1-\lambda_E)}\cdot \frac{(1-w_{SV}w_{VS})}{(1-w_{SV})}\cdot (1+w_{PP})\cdot w_{SV}
$$

when the dendrites are not inhibited in the baseline condition. Mismatch responses are caused by a release from PV neuron mediated inhibition (somatic disinhibition) that is strong enough to over-compensate for an excess of dendritic inhibition that is forwarded to the soma (when dendrites are not rectified).

# Appendix 2

We performed a mathematical analysis of a simplified model to identify the constraints that are imposed on the interneuron circuit by the presence of pPE neurons. We first describe the assumptions made and the definition of pPE neurons. We then derive the constraints for a simplified network with canonical interneuron connectivity. The solutions provide the relationship for the strength of synapses between different neuron types that must be satisfied for pPE neurons to emerge.

## Constraints for the interneuron circuit by the presence of pPE neurons

To derive the constraints for the interneuron network that are imposed by the presence of pPE neurons, we performed an analogous mathematical analysis of a simplified network model. In contrast to nPE circuits, we now assume that SOM neurons receive motor-related input, while VIP neurons receive visual input. The simplifications for the derivation rely on the following assumptions:

1. During baseline, feedback and mismatch phases, SOM interneuron-mediated inhibition exceeds excitatory motor predictions arriving at the apical dendrites of PCs.
2. Any excess of inhibition in the dendrite does not affect the soma of PCs.
3. During baseline, feedback and mismatch phases, all neuron types have positive firing rates, such that the rate rectification can be neglected.

These assumptions allow us to omit the dendritic compartment of PCs and consequently all synapses thereto. The remaining system of linear equations describes the activity of all neuron types during baseline, feedback and mismatch phase. For the subsequent analysis, we furthermore consider a homogeneous network, that is, all weights, neuronal properties and the number of incoming connections for cells of the same type are the same. As a result, we can reduce the high-dimensional system to four equations, each describing the dynamics of one representative firing rate per neuron type:

$$\tau \frac{d\mathbf{r}}{dt} = -\mathbf{r} + \Omega\mathbf{r} + \mathbf{X}, \tag{39}$$

where $\tau$ denotes the rate time constant, $\mathbf{r} = [r_E, r_P, r_S, r_V]^T$ (subscripts refer to the different neuron types; E: soma of PC, P: PV, S: SOM, V: VIP), $\Omega$ is the weight matrix and $\mathbf{X}$ denotes the external inputs. In the steady state, the firing rates are given by

$$\mathbf{r} = -(\Omega - \mathbb{1})^{-1}\mathbf{X} = W^{-1}\mathbf{X} \tag{40}$$

with the effective connectivity matrix $W$ that includes the leak:

$$W = \begin{pmatrix} -1 & -w_{EP} & 0 & 0 \\ w_{PE} & -1-w_{PP} & -w_{PS} & -w_{PV} \\ w_{SE} & 0 & -1 & -w_{SV} \\ w_{VE} & 0 & -w_{VS} & -1 \end{pmatrix}. \tag{41}$$

The weight parameters $w_{XY}$ between neuron types are strictly positive to maintain the excitatory/inhibitory nature of the various neuron types. In our model, an excitatory neuron is classified as a perfect pPE neuron, if

$$r_E^{(feedback)} = r_E^{(mismatch)} = r_E^{(baseline)}, \tag{42}$$

$$r_E^{(playback)} > r_E^{(baseline)}. \tag{43}$$

The conditions according to which no change in activity occurs in either feedback or mismatch phase (see *Equation 42*) impose constraints on the weight configuration that need to be satisfied. These can be summarized by

$$0 = W^{-1}\mathbf{X}^{fb}, \tag{44}$$

$$0 = W^{-1}\mathbf{X}^{mm}, \tag{45}$$

where $\mathbf{X}^{fb}$ and $\mathbf{X}^{mm}$ denote the excess external inputs above baseline during feedback and mismatch phase, respectively,

$$\mathbf{X}^{fb} = [V_{\mathrm{E}}, V_{\mathrm{P}} + M_{\mathrm{P}}, 1, 1]^{T} \cdot s, \tag{46}$$

$$\mathbf{X}^{mm} = [0, M_{\mathrm{P}}, 1, 0]^{T} \cdot s, \tag{47}$$

with $s$ representing a varying excitatory stimulus strength. The parameters $V_X, M_X \in \{0, 1\}$ indicate whether neuron type $X$ receives visual and motor-related inputs, respectively, and control the different input configurations.

The constraints (*Equation 44*) and (*Equation 45*) defining pPE neurons are then given by

$$0 = (1 - w_{\mathrm{SV}}w_{\mathrm{VS}})(1 + w_{\mathrm{PP}})V_{\mathrm{E}} - w_{\mathrm{EP}}(1 - w_{\mathrm{SV}}w_{\mathrm{VS}})(V_{\mathrm{P}} + M_{\mathrm{P}})$$
$$+ w_{\mathrm{EP}}w_{\mathrm{PS}}(1 - w_{\mathrm{SV}}) + w_{\mathrm{EP}}w_{\mathrm{PV}}(1 - w_{\mathrm{VS}}), \tag{48}$$
$$0 = -(1 - w_{\mathrm{SV}}w_{\mathrm{VS}})M_{\mathrm{P}} + w_{\mathrm{PS}} - w_{\mathrm{VS}}w_{\mathrm{PV}}. \tag{49}$$

These two equations yield

$$w_{\mathrm{PV}} = V_{\mathrm{P}} + w_{\mathrm{SV}}M_{\mathrm{P}} - \frac{1 + w_{\mathrm{PP}}}{w_{\mathrm{EP}}}V_{\mathrm{E}}, \tag{50}$$

$$w_{\mathrm{PS}} = w_{\mathrm{VS}}w_{\mathrm{PV}} + (1 - w_{\mathrm{SV}}w_{\mathrm{VS}})M_{\mathrm{P}}. \tag{51}$$

*Equation 50 and 51* are the mathematical formulation of an E/I balance of multiple pathways shown in *Appendix 2—figure 1*. This balance can be learned by experience-dependent inhibitory plasticity (*Appendix 2—figure 2*).

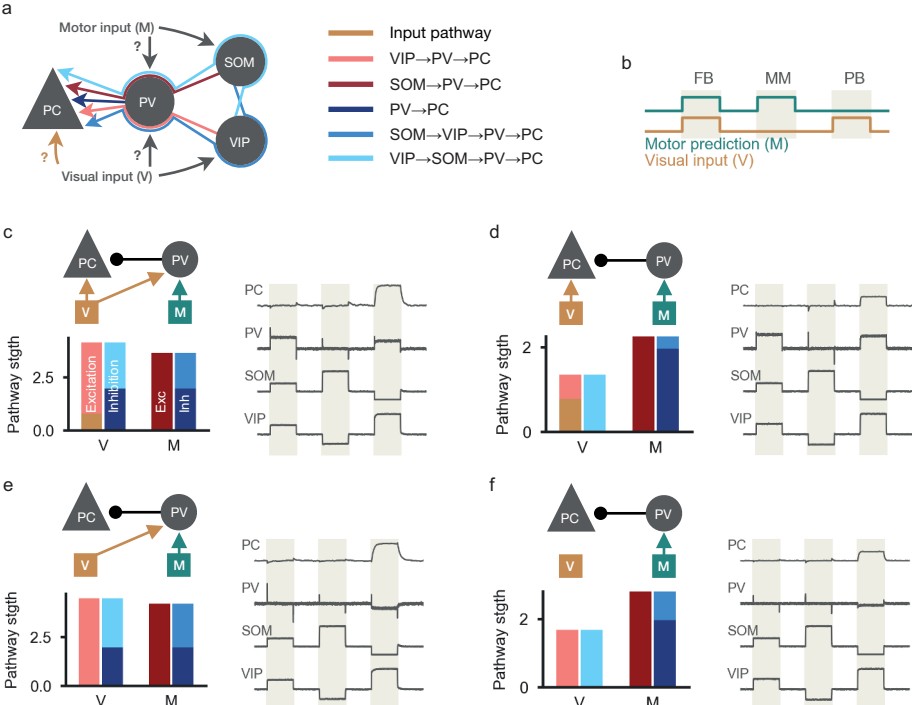

**Appendix 2—figure 1.** Multi-pathway balance of excitation and inhibition in different pPE neuron circuits. (**a**) Excitatory, inhibitory, disinhibitory and dis-disinhibitory pathways onto PCs that need to be balanced in pPE neuron circuits. Input to the soma of PCs and PV neurons is varied (**c–f**). VIP neurons receive visual input, SOM neurons receive a motor-related prediction thereof. (**b**) Test stimuli: Feedback (FB), mismatch (MM) and playback (PB) phases of 1 s each. (**c**) PCs receive visual input. PV neurons receive visual and motor inputs (left, top). When all visual (V) and motor (M) pathways are balanced (left, bottom), PCs act as pPE neurons (right). PV neuron activity increases in

both feedback and playback phases but remains at baseline during mismatch. Responses normalized between −1 and 1 such that baseline is zero. (**d**) Same as in (**c**) but PV neurons receive motor predictions only. (**e**) Same as in (**c**) but PCs receive no visual input. PV neurons remain at baseline in the absence of visual input to the soma of PCs during feedback and mismatch. (**f**) Same as in (**c**) but PCs receive no visual input and PV neurons receive motor predictions only. PV neurons remain at baseline in the absence of visual input to the soma of PCs during feedback and mismatch.

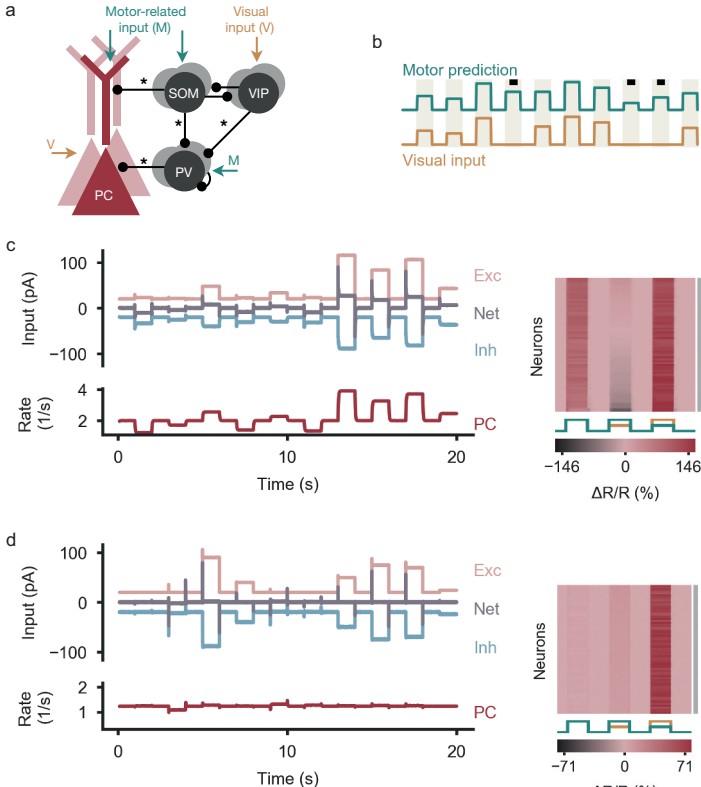

**Appendix 2—figure 2.** Balancing excitation and inhibition gives rise to positive prediction-error neurons. (**a**) Network model with excitatory PCs and inhibitory PV, SOM and VIP neurons. Connections from PCs not shown for the sake of clarity. Somatic compartment of PCs and VIP receive visual input, apical dendrites of PCs, SOM and PV neurons receive a motor-related prediction thereof. Connections marked with an asterisk undergo experience-dependent plasticity. (**b**) During plasticity, the network is exposed to a sequence of feedback (coupled sensorimotor experience) and mismatch phases (black square, no visual flow despite motor-related predictions). Stimuli last for 1 s and are alternated with baseline phases (absence of visual input and motor predictions). (**c**) Left: Before plasticity, somatic excitation (light red) and inhibition (light blue) in PCs are not balanced. Excitatory and inhibitory currents shifted by ±20 pA for visualization. The varying net excitatory current (gray) causes the PC population rate to deviate from baseline. Right: Response relative to baseline ($\Delta R/R$) of all PCs in feedback (FB), mismatch (MM) and playback (PB) phase, sorted by amplitude of mismatch response. None of the PCs are classified as pPE neurons. (**d**) Same as in (**c**) after plasticity. Somatic excitation and inhibition are balanced. PC population rate fluctuates around baseline. All PCs classified as pPE neurons. Connection strength from VIP onto SOM neurons is set to 0.8.

Please note that in the present network model, the increase during playback is caused by SOM neurons being rectified and/or a non-zero motor-related input (weaker than the visual input).

