## [Decision Letter]

**Acceptance summary:**

This manuscript explores an excitatory-inhibitory network model to determine the circuit and plasticity mechanisms underlying the generation of neurons coding for prediction errors in the visual cortex. The manuscript demonstrates that negative prediction errors arise naturally if synaptic plasticity acts to produce homeostatic excitation-inhibition balance when the circuit receives both visual inputs and internal predictions thereof. Remarkably, the emergence of prediction error neurons depends on experience of the circuit with coupled visuomotor input, reflecting recent experimental results, and the model provides direction for future optogenetic experiments to disentangle which cell types are targeted by visual or motor input.

**Decision letter after peer review:**

Thank you for submitting your article "Learning prediction error neurons in a canonical interneuron circuit" for consideration by *eLife*. Your article has been reviewed by three peer reviewers, including Srdjan Ostojic as the Reviewing Editor, and the evaluation has been overseen by Richard Ivry as the Senior Editor.

The reviewers have discussed the reviews with one another and the Reviewing Editor has drafted this decision to help you prepare a revised submission.

This manuscript models a canonical cortical microcircuit in order to investigate the circuit configurations and plasticity underlying one type of sensorimotor mismatch response identified in visual cortex. The authors show that negative prediction error neurons emerge from the network via plasticity in a set of inhibitory synapses that acts to minimize evoked firing of pyramidal cells. Moreover, the emergence of these response properties is robust to the particular cellular targets of visual and motor input (for PVs and PCs), while the type of prediction error neuron that emerges (positive versus negative) is dependent on how motor and visual inputs target VIPs and SOMs. Very nicely, the emergence of prediction error neurons depends on experience of the circuit with coupled visuomotor input (reflecting recent experimental results), and the model provides direction for future optogenetic experiments to disentangle which cell types are targeted by visual or motor input.

The general topic of microcircuits for predictive coding is very interesting, in direct relationship with experimental work. The manuscript is well written and thorough, and the topic is timely. All reviewers were generally supportive, but some concerns were raised and need to be addressed in the revision.

Essential revisions:

1) The paper focuses on a very homogeneous coding of negative prediction errors, while positive prediction errors appear only in one figure. The authors don't justify their focus on the former over the latter. It would therefore be important to include better the variety of prediction error coding in the model, and discuss its functional implications. Wouldn't it be possible and interesting to compare the distribution of different types of neurons (nPE, pPE, others) between the model and the data? The manuscript states that "nPE neurons represent only a small fraction of neurons in mouse V1". This undercuts somewhat the previous sections, and calls for a more quantitative comparison, in particular with respect to a distribution that would be obtained from random wiring for instance. The predictions for optogenetic inactivations in Figure 3 are very nice, but how do they extend to the heterogeneous case with pPE and other neuron classes (Figure 4) ? Can Equations 8-9 be extended to pPE neurons?

2) In the model, every neuron receives one or two scalar signals representing a one-dimensional visual input and corresponding motor efference copy. The authors should describe how their results would generalize to a situation in which neurons have different selectivities. In particular, there are observations that receptive-field size differs across cell types. More generally, could the circuit the authors describe generalize to neurons with "mixed selectivity" to multiple variables?

3) The paper heavily focuses on a 4 cell-type motif (Pyr, PV, SOM, VIP), but does not provide a clear message on what the functional importance of this motif is. Could PE and nPE neurons emerge in circuits with only 2 or 3 cell types? If so, what computational benefits does the presence of other cell types provide? Along similar lines, how stringent are the conditions on synaptic weights to generate nPE (or pPE) neurons ? A comparison between the number of constraints and the number of variables would be interesting to determine how large the space of solutions is.

4) What is the role of recurrent connections In the model? The argument in terms of pathways (Figure 2) does not take loops into account. The approach taken here should be compared with predictive coding of Deneve and Machens, which relies on recurrent connection.

5) A key point of the paper is the plasticity rule operative at SOM->PV synapses. In the first part of the paper, the authors use a non-local, backpropagation-like rule, and then later show that under certain circumstances this assumption can be relaxed. The fact that the initial learning rule is based on backpropagation becomes apparent only in the first paragraph of the Results, and the reader needs to dig through the Materials and methods and appendix to understand the details of the plasticity rule that is being used throughout the main text. The learning rules need to be explained earlier in the manuscript, e.g. by including, Equations 16, 17, and 19 of the Materials and methods in the main text. We also suggest reorganising the corresponding part of the Materials and methods to clearly separate the two different types of learning used at different points in the Results. More generally, during the discussion of biological plausible plasticity, the reader is left asking why the authors didn't restrict to such plasticity from the beginning. Are the results worsened when using more realistic rules?

---

## [Author Response]

Essential revisions:1) The paper focuses on a very homogeneous coding of negative prediction errors, while positive prediction errors appear only in one figure. The authors don't justify their focus on the former over the latter. It would therefore be important to include better the variety of prediction error coding in the model, and discuss its functional implications. Wouldn't it be possible and interesting to compare the distribution of different types of neurons (nPE, pPE, others) between the model and the data? The manuscript states that "nPE neurons represent only a small fraction of neurons in mouse V1". This undercuts somewhat the previous sections, and calls for a more quantitative comparison, in particular with respect to a distribution that would be obtained from random wiring for instance. The predictions for optogenetic inactivations in Figure 3 are very nice, but how do they extend to the heterogeneous case with pPE and other neuron classes (Figure 4) ? Can Equations 8-9 be extended to pPE neurons?“The paper focuses on a very homogeneous coding of negative prediction errors, while positive prediction errors appear only in one figure. The authors don't justify their focus on the former over the latter.”

We do agree that our focus on nPE neurons should be justified more clearly in the main text. There are several reasons why we decided to mainly look at nPE neurons: First of all, the study was motivated and inspired by experimental work on mismatch neurons (aka nPE neurons) that we wanted to compare our results with. Secondly, the vast majority of experimental findings on PE neurons is for nPE neurons (to the best of our knowledge). Thirdly, we assume that in the given context of visuomotor experience, nPE neurons should be the primary PE neurons. This is because we assume that an animal naturally learns over time that when it moves forward the world moves backwards (visual flow changes). At the same time the visual input can also change when the animal doesn’t move (e.g. caused by the movements of other animals in their environment). In contrast, phases in which self-movement does not cause changes in visual input are true mismatches and PE neurons should indicate this. As we assume that the experience a network is exposed to determines the nature of the PE neuron, this would indicate that the majority of PE neurons are indeed nPE neurons (in this simplified setting we studied).

To emphasize our choice, we have changed the beginning of the Results section:

“We studied a rate-based network model of layer 2/3 of rodent V1 to investigate how prediction-error (PE) neurons develop. In the following, we will focus primarily on negative prediction-error (nPE) neurons. […] However, the same approaches and principles derived for nPE neurons can also be applied to positive prediction-error (pPE) neurons.”

and the Discussion:

“Here, we have mainly focused on the development of nPE neurons because those have been studied extensively in layer 2/3 of rodent V1, which allowed us to qualitatively compare our model with experimental findings. […] Moreover, as we assume that excitatory neurons aim to establish an E/I balance for all stimuli they are regularly exposed to, and as animals experience episodes, in which the change of visual input is only caused by external factors (playback phases), excitatory neurons are more likely to develop into nPE than pPE neurons in the sensorimotor paradigm used here.”

“Wouldn't it be possible and interesting to compare the distribution of different types of neurons (nPE, pPE, others) between the model and the data? Can Equations 8-9 be extended to pPE neurons?”

Yes, it would indeed be interesting to compare the constraints for nPE and pPE neurons. Even though we assume that nPE neurons should play the major role in the visuomotor paradigm we study, the existence of pPE neurons in other contexts (e.g., reward coding) is undenied. Therefore, it would be interesting to formally derive and compare constraints on the interneuron circuit. We therefore now include mathematical constraints for pPE neurons (similar to Equation 8 and 9, see Appendix 2) and first simulation results for different pPE circuits (analog to Figure 2, see Appendix 2—figure 1). We also show how those can be learned with the same learning rules used for nPE neurons (similar to Figure 1, see Appendix 2—figure 2). Note that these results are for a homogeneous network of pPE neurons. The results, which are shown in the Appendix, are referred to in the Discussion:

“We expect that the same principles and approaches described here also hold for the formation of pPE neurons. […] The inhibitory plasticity establishes pPE neurons independent of the input configuration onto PCs and PV neurons as long as various excitatory, inhibitory, disinhibitory and dis-disinhibitory pathways can be balanced (see Appendix 2—figure 1, Equations 50 and 51)”.

A thorough and complete investigation of nPE and pPE neurons developing in the same network is beyond the scope of this study and currently an ongoing project in the lab. To understand why the formation of both PE neurons in the same network is not a trivial task in the framework we use, please consider these two cases:

i) Perfect PE neurons, that is, nPE neurons remain at baseline during feedback and playback, while pPE neurons remain at baseline during feedback and mismatch phase:

For perfect nPE neurons to develop, the network must be exposed to feedback and playback phases. For perfect pPE neurons to develop, the network must be exposed to feedback and mismatch phases during training. Thus, if both PE neurons should emerge within the very same network, the network must be exposed to all phases. However, given the current plasticity rules we use in the framework, this would most likely produce neurons that remain at their baseline during all phases. To force the network to generate nPE and pPE neurons, the plasticity rules must incorporate gating signals that restrict learning to a subset of these phases. This is beyond the current framework.

ii) Imperfect PE neurons, that is, nPE neurons remain at baseline during feedback, but are allowed to decrease activity during playback, and similarly pPE neurons remain at baseline during feedback but are allowed to decrease activity during mismatch:

In such a scenario, the network must only be exposed to feedback phases, to avoid that the plasticity establishes an E/I balance also during playback and mismatch phases. nPE and pPE neurons can then develop within the same network, but the outcome will depend on many parameters, including the network initialisation, the number of plastic synapses and the distribution of actual and expected signals. The number of degrees of freedom in the model is then very large. A complete characterisation of all parameter boundaries is subject to ongoing research in the lab and beyond the scope of the present paper.

We have now included a section on this topic in the Discussion:

“In the present work, we derived the constraints for separate nPE and pPE neurons and did not study the parallel development of both in the same neural network. […] A thorough investigation of these scenarios for the simultaneous development of nPE and pPE neurons is, however, beyond the scope of the present study.”

“The predictions for optogenetic inactivations in Figure 3 are very nice, but how do they extend to the heterogeneous case with pPE and other neuron classes (Figure 4)?”

We agree that it would be great to extend the simulated optogenetic experience to heterogeneous networks. However, as discussed above, a thorough understanding of heterogeneous networks with nPE and pPE neurons is beyond the scope of the present work. In fact, Loreen Hertäg is currently preparing a grant proposal, in which she aims to extend on this work, including an extensive study of heterogeneous networks.

2) In the model, every neuron receives one or two scalar signals representing a one-dimensional visual input and corresponding motor efference copy. The authors should describe how their results would generalize to a situation in which neurons have different selectivities. In particular, there are observations that receptive-field size differs across cell types. More generally, could the circuit the authors describe generalize to neurons with "mixed selectivity" to multiple variables?

That is a very good point and one of the foci of the above mentioned grant proposal. Indeed, it has been found that layer 2/3 mismatch neurons in rodent V1 exhibit receptive fields for mismatches, which co-align with their visual receptive fields (Zmarz and Keller, 2016). These sensorimotor mismatch responses are confined to local regions of the visual field. It is therefore very interesting to equip excitatory neurons with feature selectivity and study how the constraints on the IN circuit change and how this affects the selectivity in IN types, which have been shown to exhibit a broad spectrum of tuning properties (Sohya et al., 2007; Cardin et al., 2007; Kerlin et al., 2010; Atallah et al., 2012). We expect that nPE (and also pPE neurons) can still emerge in these networks, but further constraints will need to be satisfied. In a first approach, feature selectivity can be studied in independent columns (see e.g. Wacongne et al., 2012) with weak cross-coupling that is increased systematically. We intend to study this in detail but would prefer to leave this for future work because it will open up a completely new chapter. Nevertheless, we have included a section on this topic in the Discussion:

“A hallmark of neurons in sensory areas is their pronounced feature selectivity (Cardin et al., 2007; Niell and Stryker, 2008; Harris and Mrsic-Flogel, 2013). […] For future work, it would be interesting to study how the presence of feature-selective PE neurons constrains the feature selectivity in interneurons that tend to be more broadly tuned than excitatory neurons (Sohya et al., 2007; Cardin et al., 2007; Kerlin et al., 2010; Atallah et al., 2012).”

3) The paper heavily focuses on a 4 cell-type motif (Pyr, PV, SOM, VIP), but does not provide a clear message on what the functional importance of this motif is. Could PE and nPE neurons emerge in circuits with only 2 or 3 cell types? If so, what computational benefits does the presence of other cell types provide? Along similar lines, how stringent are the conditions on synaptic weights to generate nPE (or pPE) neurons ? A comparison between the number of constraints and the number of variables would be interesting to determine how large the space of solutions is.

Thanks for this suggestion. We decided to focus on these three IN types because they form a canonical motif that has been observed frequently. However, we agree that a more extensive discussion further strengthens the paper. We therefore now include an exhaustive list of nPE circuits and the corresponding constraints in the Appendix that hopefully clarifies this matter. Moreover, we adapted the text to account for this:

“We used a mathematical analysis to derive constraints imposed on an interneuron circuit by the presence of nPE neurons. […] While a minimal model that allows nPE neurons to develop comprises SOM neurons and PCs (Attinger et al., 2017), the network with three inhibitory neuron types appears the most likely nPE circuit given what is currently known about rodent V1.”

4) What is the role of recurrent connections In the model? The argument in terms of pathways (Figure 2) does not take loops into account. The approach taken here should be compared with predictive coding of Deneve and Machens, which relies on recurrent connection.

Thanks for this comment. It highlights that we have not sufficiently emphasized that the model in fact includes recurrent connections, not only between the PCs and the INs, but also within the PC population. To keep the figures clean, excitatory connections are not includes in the network schemata. To highlight that those are included, we updated the corresponding captions. Note that during feedback and playback, these connections are not relevant because the E-E synapses target the dendrites, which are over-inhibited in those phases. Please note also that the generality of the approach does not rely on the recurrent connections targeting the dendrites.

Thanks for highlighting that the related work of Deneve and Machens should be discussed in the paper. We have now added a section in the Discussion:

“In our model, the excitatory recurrent connections target the apical dendrites of PCs, but given that PCs comprise a homogeneous population, they serve no specific computational purpose in the present context. […] It will be interesting to study how this line of work is related to the PE circuit model we studied here, but it would require to extend the present model to perform richer computations, e.g., by endowing it with stimulus selectivity.”

5) A key point of the paper is the plasticity rule operative at SOM->PV synapses. In the first part of the paper, the authors use a non-local, backpropagation-like rule, and then later show that under certain circumstances this assumption can be relaxed. The fact that the initial learning rule is based on backpropagation becomes apparent only in the first paragraph of the Results, and the reader needs to dig through the Materials and methods and appendix to understand the details of the plasticity rule that is being used throughout the main text. The learning rules need to be explained earlier in the manuscript, e.g. by including, Equations 16, 17, and 19 of the Materials and methods in the main text. We also suggest reorganising the corresponding part of the Materials and methods to clearly separate the two different types of learning used at different points in the Results. More generally, during the discussion of biological plausible plasticity, the reader is left asking why the authors didn't restrict to such plasticity from the beginning. Are the results worsened when using more realistic rules?

Thanks for pointing out that the plasticity rules used are not obvious to the reader and should therefore be mentioned earlier. We now describe the plasticity rules in the Results (second paragraph and subsection “nPE circuits can also be learned by biologically plausible learning rules” and reference to the respective equations in the Materials and methods more carefully. The Materials and methods now includes subsections that hopefully make the distinction between the two different rules more visible (subsections “Connections onto PCs”, “Connections onto PV neurons - non-local learning” and “Connections onto PV neurons - local approximation to backpropagation of error”).

Regarding the performance of the local approximation of the backprop rule:

The main goal of the present paper was to show that PE circuits can be learned by balancing excitation and inhibition, rather than on the specific synaptic learning rules that achieve an E/I balance. In our setting, the local, more biological learning rule does a very good job (see Figure 6). We also found that this local rule effectively establishes an E/I balance in networks with rich stimulus selectivity (Mackwood et al., 2020). Of course, we expect that the approximation will break down at some point, e.g., when the connection probability is too low. An extensive evaluation of the conditions under which the approximation works would in our view distract from the main message of the paper and should be done elsewhere.

To justify our choice, we added a sentence in the Discussion:

“Given that the main goal of the present paper was to show that PE circuits can be learned by balancing excitation and inhibition, we used the plasticity rule implementing a backpropagation of error, to ensure maximal generality.”